# Characterization and Learning of Causal Graphs with Latent Confounders and Post-treatment Selection from Interventional Data

**Gongxu Luo**[1], **Loka Li**[1], **Guangyi Chen**[1,2], **Haoyue Dai**[2], **Kun Zhang**[1,2]
[1] Mohamed bin Zayed University of Artificial Intelligence, [2] Carnegie Mellon University
{gongxu.luo, kun.zhang}@mbzuai.ac.ae

## Abstract

Interventional causal discovery seeks to identify causal relations by leveraging distributional changes introduced by interventions, even in the presence of latent confounders. Beyond the spurious dependencies induced by latent confounders, we highlight a common yet often overlooked challenge in the problem due to post-treatment selection, in which samples are selectively included in datasets after interventions. This fundamental challenge widely exists in biological studies; for example, in gene expression analysis, both observational and interventional samples are retained only if they meet quality control criteria (e.g., highly active cells). Neglecting post-treatment selection may introduce spurious dependencies and distributional changes under interventions, which can mimic causal responses, thereby distorting causal discovery results and challenging existing causal formulations. To address this, we introduce a novel causal formulation that explicitly models post-treatment selection and reveals how its differential reactions to interventions can distinguish causal relations from selection patterns, allowing us to go beyond traditional equivalence classes toward the underlying true causal structure. We then characterize its Markov properties and propose a $\mathcal{F}$ine-grained $\mathcal{I}$nterventional equivalence class, named $\mathcal{FI}$-Markov equivalence, represented by a new graphical diagram, $\mathcal{F}$-PAG. Finally, we develop a provably sound and complete algorithm, $\mathcal{F}$-FCI, to identify causal relations, latent confounders, and post-treatment selection up to $\mathcal{FI}$-Markov equivalence, using both observational and interventional data. Experimental results on synthetic and real-world datasets demonstrate that our method recovers causal relations despite the presence of both selection and latent confounders.

## 1 Introduction

Causal discovery from interventional (and observational) data, often referred to as interventional causal discovery, aims to identify causal relations by exploiting distributional changes induced by interventions (Spirtes et al., 2000; Pearl, 2000). Despite progress in interventional causal discovery in handling latent confounders, pre-treatment selection (Dai et al., 2025), and biological constraints (Luo et al., 2025), we highlight a common yet often overlooked problem, *post-treatment selection*, which refers to the selective inclusion of samples after interventions (Heckman, 1978). For example, in gene perturbation studies, only perturbed cells (intervention) that pass the quality control (selection) are profiled (Norman et al., 2019). In Clinical Trial Per-Protocol Analysis, only participants completing over 80% of scheduled visits (e.g., up to week 12) are included in the final analysis (Detry & Lewis, 2014). Failure to account for post-treatment selection introduces spurious dependencies and intervention-driven distributional changes that mimic causal responses, thereby leading to incorrect statistical inference and challenging existing interventional causal formulations.

Specifically, existing interventional formulations neither distinguish causal relations from post-treatment selection nor detect where the selection is present. Mainstream frameworks identify causal relations and characterize interventional Markov equivalence classes (ECs) on DAGs by exploiting a cross-intervention pattern: after interventions on the cause, marginal distribution $p(\text{effect})$ changes, and conditional distribution $p(\text{effect}|\text{cause})$ remains (Tian & Pearl, 2001; Hauser & Bühlmann, 2012; 2015). When latent confounders are present, variations in $p(\text{effect}|\text{cause})$ are further utilized to

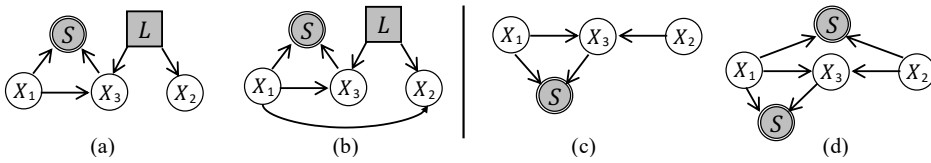

Figure 1: Motivation examples. (a) & (b) exhibit same dependence with tails from $X_1$ and arrowheads into $X_2$, regardless of direct causation; (c) & (d) exhibit same dependence with tails on both $X_1$ and $X_2$, regardless of direct selection. Existing methods cannot distinguish these cases, whereas ours can.

characterize the interventional ECs involving latent confounders (Eaton & Murphy, 2007; Ghassami et al., 2017; Kocaoglu et al., 2019; Zhou et al., 2025). However, post-treatment selection is non-identifiable within these frameworks because it yields the same pattern, variant $p(\text{effect})$ and invariant $p(\text{effect}|\text{cause})$ before and after the intervention, as causal relations. For example, under post-treatment selection, Figure 1(a) exhibits the same pattern (variant $p(X_2)$ and invariant $p(X_2 \mid X_1)$ after intervening on $X_1$) as (b). Subsequently, current frameworks place (a) and (b) in the same EC (same representation), regardless of whether a direct causal link exists between $X_1$ and $X_2$, thereby failing to identify causal relations from post-treatment selection. An analogous non-identifiability arises for direct selection, as illustrated in Figure 1(c) and (d). This representational gap challenges existing frameworks and motivates a new formulation that explicitly models post-treatment selection.

In this paper, we examine the causal structure among intervened variables in the general setting involving latent confounders and selection bias, explicitly handling post-treatment selection without imposing graphical or parametric assumptions. First, we demonstrate how causal relations, latent confounders, and post-treatment selection differ in structural symmetries (e.g., selection structure with both tails on endpoints, while causation is not) and distributions (variability and invariance) after intervention, which are characterized by conditional independence (CI) patterns. Second, building on these observations, we propose a $\mathcal{F}$ine-grained $\mathcal{I}$nterventional equivalence class (e.g., distinguishing Figure 1(a) from (b), and (c) from (d)), named $\mathcal{FI}$-Markov equivalence, and provide detailed characterizations. In graphical representation, partial ancestral graph (PAG) edges encode ECs with a broad range of possible structures and thus prevent the unique graphical representation. To obtain a more concise and expressive graphical representation for $\mathcal{FI}$-Markov equivalence, we introduce $\mathcal{F}$-PAG, an extension of the PAG diagram that incorporates novel edge types. Third, we present a sound and complete algorithm $\mathcal{F}$-FCI for recovering the $\mathcal{FI}$-Markov equivalence class.

**Contributions.** In this paper, we focus on a fundamental yet largely overlooked problem, the *post-treatment selection* that lies beyond the scope of existing interventional causal discovery frameworks. First, we introduce a causal formulation that models post-treatment selection in the presence of latent confounders, and we define the novel $\mathcal{FI}$-Markov equivalence and $\mathcal{F}$-PAG accordingly. Second, building on this formulation, we develop a new algorithm $\mathcal{F}$-FCI, which integrates intervention-based CI patterns with tailored orientation rules. Theoretically, we prove its soundness and completeness. Third, we validate our approach on both synthetic and real-world datasets, demonstrating its effectiveness. Collectively, these contributions provide a principled framework for distinguishing post-treatment selection from true causal relations, thereby broadening the scope of interventional causal discovery.

## 2 PRELIMINARIES AND MOTIVATION

In this section, we first introduce the graphical causal model that involves both latent confounders and selection bias (§ 2.1). We then review the standard paradigm for interventional causal discovery (details in Appendix C) and demonstrate why it fails to handle post-treatment selection (§ 2.2).

### 2.1 GRAPHICAL CAUSAL MODEL WITH LATENT CONFOUNDERS AND SELECTION BIAS

We begin with the general problem setup: a DAG with latent confounders and selection bias. Let the DAG $\mathcal{G}$ on vertices with index $[N] := \{1, \cdots, N\}$ encode the structure of the underlying causal model where vertices correspond to observed random variables $X = (X_i)_{i=1}^N$. For any subset $A \subset [N]$, let $X_A := (X_i)_{i \in A}$ and by convention $X_\varnothing \equiv 0$. Apart from the observed ones, $L = \{L_i\}_{i=1}^R$ accounts for the confounders that affect $X$ but remain unobserved (latent confounders), and the exogenous selection variable $S = \{S_i\}_{i=1}^T$ generally represent both *pre-treatment selection* (preferential inclusion of samples before intervention) and *post-treatment selection* (arising after intervention) (Heckman, 1978; Elwert & Winship, 2014). In this paper, we specialize in

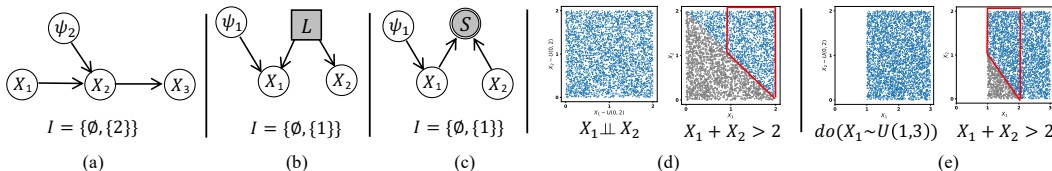

Figure 2: Examples of graphical representations. (a) Augmented DAG with explicit intervention indicators ($\psi$). (b) Extension of the augmented DAG to include latent confounders. (c) Modeling post-treatment selection using the augmented DAG, with toy examples of selection on observational data (d) and selection after intervention (e), where the positively invariant $p(X_2|X_1)$ is marked in red.

post-treatment selection and assume selection works on at least two observed variables. Throughout, analyses are conducted conditional on $S = 1$ (i.e., within the selected sample).

To represent the general graph with latent confounders and selection bias, the ancestral graph is defined by a mixed graph without direct and indirect cycles (detailed in Definition 8). To investigate the learnability of graphical models and the information-theoretic limits of the CI test on observational data, the Markov properties of ancestral graphs are examined. Analogous to the d-separation (Definition 7) criterion used for DAGs, the m-separation (Definition 9) blocks the paths in ancestral graphs. Under the pairwise Markov property, in which the absence of edges reflects conditional independence, the ancestral graph that satisfies this property, a.k.a. maximality assumption (Definition 10), is the Maximal Ancestral Graph (MAG). Given that the global, local, and pairwise Markov properties enable the recovery of causal structures via CI tests, under the representation of the MAG diagram, different graphs that entail the same m-separation form the Markov equivalence class (Definition 11).

## 2.2 LIMITATIONS OF EXISTING INTERVENTIONAL CAUSAL DISCOVERY PARADIGM UNDER POST-TREATMENT SELECTION

Interventional causal discovery aims to learn the structure of $\mathcal{G}$ from data collected under multiple (hard and soft) intervention settings, each with an *intervention target* $I \subset [N]$, meaning variables $X_I$ are intervened on. Let $\mathcal{I} = \{I^{(0)}, I^{(1)}, \ldots, I^{(K)}\}$ denote the collection of intervention targets across $K$ interventions, and $\{p^{(0)}, p^{(1)}, \ldots, p^{(K)}\}$ indicate the corresponding *interventional distributions* over $X$. We assume throughout $I^{(0)} = \varnothing$, i.e., the pure observational data is available.

Hard interventions remove all incoming edges to the vertices in the intervention target $I^{(k)}$ in $\mathcal{G}$ while all other edges remain. Soft interventions do not break any arrows incident on the intervention target; instead, they only change the conditional distribution (Eberhardt & Scheines, 2007). Rather than analyzing each interventional setting separately, a more effective approach is to exploit changes and invariances across settings: intervening on a cause alters the marginal distribution $p(\text{effect})$, while the conditional $p(\text{effect}|\text{cause})$ remains invariant. Conversely, intervening on an effect leaves $p(\text{cause})$ unchanged, but $p(\text{cause}|\text{effect})$ changes (Hoover, 1990; Tian & Pearl, 2001). Such invariance is exploited in the *invariance causal inference framework* (Meinshausen et al., 2016; Ghassami et al., 2017) and has been extended to settings with latent confounders (Jaber et al., 2020).

To formally exploit such invariance analysis and model "the action of changing targets", Newey & Powell (2003); Korb et al. (2004) introduce the *augmented DAG*, denoted by $\text{aug}(\mathcal{G}, \mathcal{I})$, which, as shown in Figure 2(a), extends the original $\mathcal{G}$ by adding exogenous binary vertices $\psi = \{\psi_{I^{(k)}}\}_{k=1}^{K}$ as *intervention indicators*, each pointing to its target $I^{(k)}$. Whether the $k$-th intervention alters a marginal density, i.e., $p^{(0)}(X_A) \neq p^{(k)}(X_A)$ or equivalently $p(X_A \mid \psi_{I^{(k)}} = 0) \neq p(X_A \mid \psi_{I^{(k)}} = 1)$, is then nonparametrically represented by the CI relation $\psi_{I^{(k)}} \not\perp\!\!\!\perp X_A$, and graphically represented by the *d-separation* $\psi_{I^{(k)}} \not\perp\!\!\!\perp_d X_A$ in $\text{aug}(\mathcal{G}, \mathcal{I})$, where $\perp\!\!\!\perp_d$ denotes d-separation and $\not\perp\!\!\!\perp_d$ d-connection. Moreover, latent confounders can also be incorporated into augmented DAGs, as shown in Figure 2(b). The invariance in marginal distributions ($p^{(0)}(X_2) = p^{(1)}(X_2)$) represented by $\psi_1 \perp\!\!\!\perp X_2$ and variability in conditional distributions ($p^{(0)}(X_2|X_1) \neq p^{(1)}(X_2|X_1)$) represented by $\psi_1 \not\perp\!\!\!\perp X_2|X_1$) after intervention help distinguish latent confounders from causal relations. Causal discovery algorithms like PC (Spirtes et al., 2000) and FCI (Spirtes et al., 2000; Zhang, 2008b) have been applied in this context (Zhang, 2008a; Huang et al., 2020; Magliacane et al., 2016; Kocaoglu et al., 2019), and *augmented MAG* have been developed as the corresponding graphical representation.

Building on the established framework of interventional causal discovery, when selection appears after intervention, the post-treatment selection induces changes in the marginal distribution $p(\text{effect})$ while keeping invariant in the conditional distribution $p(\text{effect}|\text{cause})$, as illustrated in Figure 2(c): $p^{(1)}(X_2 \mid S = 1) \neq p^{(0)}(X_2 \mid S = 1)$ and $p^{(1)}(X_2 \mid X_1, S = 1) = p^{(0)}(X_2 \mid X_1, S = 1)$ with examples in (d) and (e). Although post-treatment selection can be represented within the augmented framework, its invariant and variant characteristics are indistinguishable from those of causal relations, rendering it non-identifiable as discussed in Figure 1. This motivates a new formulation that models post-treatment selection and identifies true causal relations.

## 3 NEW FORMULATION: INTERVENTION MEETS POST-TREATMENT SELECTION

Based on the exploration of the interventional causal discovery paradigm, in this section, we extend the paradigm to design a new formulation for post-treatment selection in the presence of latent confounders (§ 3.1), characterize the Markov properties (§ 3.2), and provide the graphical criteria for determining whether two augmented DAGs are Markov equivalent given the same interventions (§ 3.3).

### 3.1 MODELING POST-TREATMENT SELECTION

The first step is to model the post-treatment selection explicitly. Since the variant and invariant characteristics are consistent with the Markov assumption, post-treatment selection can be naturally modeled within the augmented DAG (see § 2.2) by adding a selection variable $S$. Accordingly, we adopt the augmented DAG to coherently unify observational and interventional data by introducing an intervention indicator $\psi$. Under this model, the joint distribution over the observed variables $X$ in the $k$-th intervention, conditioning on post-treatment selection denoted by $p_s^{(k)}(X)$, factorizes as

$$p_s^{(k)}(X) = \prod_{\{i|\{i\}\subset I^{(k)}\}} p^{(k)}(X_i|\hat{X}_{pa_{\mathcal{G}}(i)}, S = 1) \prod_{\{j|\{j\}\not\subset I^{(k)}\}} p^{(0)}(X_j|\hat{X}_{pa_{\mathcal{G}}(j)}, S = 1), \quad (1)$$

where $\hat{X}_{pa_{\mathcal{G}}(i)} \subset X \cup L$ indicates the parents of $X_i$, and $S = 1$ indicates the presence of post-treatment selection. To represent the details of the structural causal model, the graph involving $S$ and $L$ is represented by the augmented DAG, which is redefined as follows (details in Definition 12).

**Definition 1 (Augmented DAG).** For a DAG $\mathcal{G}$ over $X \cup L \cup S$ and intervention target $I \subset [N]$, the *augmented graph* $Aug_{\mathcal{I}}(\mathcal{G})$ is a DAG with vertices $\psi \cup X \cup L \cup S \cup \epsilon$, where: $\psi = \{\psi_{I^{(k)}}\}_{k=1}^{K}$ is a set of exogenous binary indicators for the representation of marginal changes between two environments (observation-intervention or intervention-intervention), pointing to the corresponding intervened variable $X_{I^{(k)}}$; $\epsilon$ is exogenous noise for variables, whether observed or hidden.

An illustration of the augmented DAG is shown in Figure 3, depicting the data generation process. In $Aug_{\mathcal{I}}(\mathcal{G})$, only $X$ and $\psi$ are measurable, forming the basis of the representation of different environments, such as observational data $p(X|\psi = 0, S = \mathbf{1})$ and interventional data $p(X|\psi = 1, S = \mathbf{1})$. $S = \mathbf{1}$ is conditioned on, meaning all individuals, whether observed or intervened, are selected at

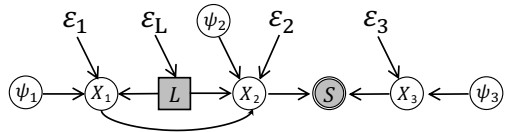

Figure 3: Illustration of a structural causal model (SCM) represented by an augmented DAG.

the outset. Moreover, following the diagram of MAG for marginalized representation (Zhang, 2008b), each Augmented DAG can also be formally represented by the corresponding Augmented MAG.

### 3.2 CHARACTERIZING MARKOV PROPERTIES

Our ultimate goal is to learn the causal structure from both observational and interventional data. On the rationale of modeling post-treatment selection and marginal changes between observational and interventional data using the Augmented DAG $Aug_{\mathcal{I}}(\mathcal{G})$ in §3.1, the standard Markov properties, i.e., the global Markov property (formulated via d-separation) and the local Markov property (each node is conditionally independent of its non-descendants given its parents), hold exactly as they do in conventional DAGs. These properties provide the theoretical foundation for using CI tests in structure learning and offer an information upper bound for the CI implementations.

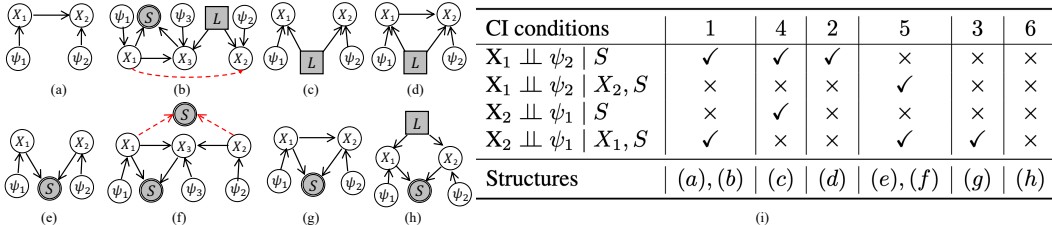

Figure 4: Characterizing Markov properties with CI patterns (i) of augmented DAGs (a)-(h). Red dashed lines indicate that CI patterns persist regardless of whether $X_1 \dashrightarrow X_2$ (b) or $X_1 \dashrightarrow S \dashleftarrow X_2$ (f), whereas $\psi_3 \not\perp\!\!\!\perp X_2$ provides evidence of the presence of a direct causal link or selection.

By leveraging the Markov properties, CI tests can recover causal structure from data. In particular, three classes of statistical signals are informative: 1) **Interventional distribution changes:** Variability in marginal observational and interventional distributions manifests as conditional dependencies between the intervention indicator $\psi$ and affected variables $X$. For example, in Figure 4(a), $\psi_1 \not\perp\!\!\!\perp X_2$ indicates that perturbing $X_1$ propagates a distributional change to $X_2$. 2) **Invariant relations:** Equality of conditional distributions across observational and interventional data signals invariance. Specifically, $\psi_1 \perp\!\!\!\perp X_2 \mid X_1$ indicates the invariance of $p^{(0)}(X_2 \mid X_1) = p^{(1)}(X_2 \mid X_1)$, $I^{(1)} = \{1\}$. 3) **Structural symmetries:** Certain structures exhibit characteristic symmetry in their CI patterns. For instance, a symmetric selection shown in Figure 4(e) yields $\psi_1 \perp\!\!\!\perp X_2 \mid X_1$, $\psi_2 \perp\!\!\!\perp X_1 \mid X_2$, $\psi_1 \not\perp\!\!\!\perp X_2$, $\psi_2 \not\perp\!\!\!\perp X_1$. Below, we formally define these relations implied by the model.

**Theorem 1** (CI and invariance implementation). *For positive interventional distributions $p^{(k)}(X)$ and observational distribution $p^{(0)}(X)$ generated from the DAG $\mathcal{G}$ in the presence of latent confounders $L$ and selection $S$ with intervention targets $\{I^{(k)}\}_{k \in \{0\} \cup [K]}$, let $\{Aug_{I^{(k)}}(\mathcal{G})\}_{k \in \{0\} \cup [K]}$ be the corresponding augmented DAGs. For any disjoint $A, B, C \in [N]$, the following statement holds:*

- *For any $k \in \{0\} \cup [K]$, if $X_A \perp\!\!\!\perp_d X_B | \{X_C, S\}$ holds in $Aug_{I^{(k)}}(\mathcal{G})$, then $X_A \perp\!\!\!\perp X_B | \{X_C, S\}$ in $p^{(k)}$.*
- *For any $k \in [K]$, if $\psi_{I^{(k)}} \perp\!\!\!\perp_d X_A | X_B$ holds in $Aug_{I^{(k)}}(\mathcal{G})$, then $p^{(k)}(X_A|X_B) = p^{(0)}(X_A|X_B)$.*
- *For any $k \in [K]$, if $\psi_{I^{(k)}} \not\perp\!\!\!\perp_d X_A \mid \varnothing$ holds in $Aug_{I^{(k)}}(\mathcal{G})$, then $p^{(k)}(X_A) \neq p^{(0)}(X_A)$.*

**Remark 1.** $\psi_{I^{(k)}}$ generally marginalizes changes between different environments. The difference between the two interventions on the same $I^{(k)}$ also follows the statements in Theorem 1, where the hard-hard intervention changes the causal diagram, providing additional information that is only used to identify the structures of unblocked paths.

Theorem 1 shows that invariance and variability in marginal and conditional distributions are implied by graphical conditions, namely d-separation among $\psi \cup X | S = 1$ in augmented DAGs. Previous studies leveraged this statistical information to distinguish causal effects from associations induced by latent confounders. However, it is known that selection bias also introduces spurious dependencies:

**Lemma 1** (Additional dependencies induced by selections). *For any DAG on $X_{[N]} \cup L \cup S$, targets $\mathcal{I} \in [N]$, and disjoint $A, B, C \in [N]$, if $X_A \perp\!\!\!\perp_d X_B|X_C, S$ holds in the augmented DAG $Aug_{\mathcal{I}}(\mathcal{G})$, then $X_A \perp\!\!\!\perp_d X_B|X_C$ holds in the original DAG. The reverse is not necessarily true.*

With the characterization of global and local Markov properties ready, differences in graphical properties, such as asymmetry, captured in CI patterns as shown in Figure 4, help distinguish different structures. Specifically, Figure 4(i) shows that (a) and (e) exhibit different CI patterns, with (e) being symmetric. We further observe that although (a) and (b) share the same CI patterns between $X_1$ and $X_2$ regardless of whether a direct causal link exists in (b), they differ in their underlying causal structures. This is due to the Y-structure at $X_2$ forming an unblocked path, which exhibits the same characteristics as causation. Beyond focusing only on cause and effect targets, hard interventions on $X_3$ open the path by blocking the selection effect on the latent confounder $L$. The variation in two different hard interventions on $X_3$ can be modeled by $\psi_3$ for representation. Then, $\psi_3 \not\perp\!\!\!\perp X_2$ allows us to distinguish case (b) and assess whether there is a direct causal link between $X_1$ and $X_2$. Similarly, direct selection can be identified in the same way for (e) and (f). This, in turn, goes beyond traditional ECs and can identify concrete causal structures at the DAG level.

### 3.3 $\mathcal{FI}$-Markov Equivalence

In § 3.2, we characterized the Markov properties implied by the true model of the data. Now, to identify the true model from data, in this section, we shall understand to what extent the true model is *identifiable*, as different models may share identical CI implications, namely, being *Markov equivalent*. To characterize the equivalence class under the general setting with MAG representation, the Markov equivalence with corresponding m-separation on observational data is discussed. However, with the help of interventional data, we propose a novel $\mathcal{F}$ine-grained $\mathcal{I}$nterventional Markov Equivalence, named $\mathcal{FI}$-Markov equivalence (Definition 2), and characterize the EC under the augmented DAG framework based on the Markov properties. Different from the graphical representation of the structural causal model, the representation of learned ECs is only over observations $X$. We subsequently extend the Partial Ancestral Graph (PAG) framework (Definition 4) with novel edges for the representation of $\mathcal{FI}$-Markov ECs, which are more informative compared with PAG.

Because the Markov property allows us to distinguish structures that existing formulations cannot, for instance, whether a direct causal link exists in Figure 4(b), we define a new $\mathcal{FI}$-Markov equivalence under the augmented DAG framework for a more precise structural representation. Two different augmented DAGs with the same intervention targets can entail the same CI patterns in the data. Formally,

**Definition 2** ($\mathcal{FI}$-**Markov equivalence**). Two Augmented DAGs, $Aug_{\mathcal{I}}(\mathcal{G}_1)$ and $Aug_{\mathcal{I}}(\mathcal{G}_2)$, are $\mathcal{FI}$-*Markov equivalent* with the same intervention targets $\mathcal{I}$, if and only if they have the same d-separation (the same skeleton and v-structure in the description of the corresponding MAG representation) among $X_{[N]\setminus\mathcal{I}}$, and the same CI patterns between $\psi$ and any intervened variable $X_i \in X_{\bigcup\mathcal{I}}$.

#### 3.3.1 Graphical criteria for $\mathcal{FI}$-Markov Equivalence

When learning the EC based on Markov properties, causal discovery methods only recover whether the unblocked paths have a tail or an arrowhead at each endpoint over $X_{[N]}$, not the full structure with $L$ and $S$ (Kocaoglu et al., 2017). These unblocked paths are named inducing paths, defined as follows:

**Definition 3** (**Inducing path**). In augmented DAGs, $X_i$, $X_j$ are any two vertices, and $L$, $S$ are disjoint sets of vertices not containing $X_i$, $X_j$. A path $p$ between $X_i$, $X_j$ is called an inducing path relative to $\langle L, S \rangle$ if every non-endpoint vertex on $p$ is either in $L$ or a collider, and every collider on $p$ is an ancestor of either $X_i$, $X_j$, or a member of $S$.

**Example.** In Figure 4(b), the path between $X_1$ and $X_2$ is the inducing path with unblocked $X_3$.

To characterize the ECs via graphical features over $X_{[N]}$, we still need to borrow the MAG representation for marginalization. Following the procedure of MAG construction introduced in (Zhang, 2008b), every augmented DAG corresponds to an augmented MAG $\mathcal{M}(Aug_I(\mathcal{G}))$ in graphical representation. Then, we show the rules to construct $\mathcal{M}(Aug_I(\mathcal{G}))$ by presenting the following lemmas.

**Lemma 2** (When are two variables dependent in observational data?). *For any $i, j \in [N]$, $X_i$ and $X_j$ are adjacent in $\mathcal{M}(Aug_I(\mathcal{G}))$, if and only if $X_i$ and $X_j$ have at least one inducing path in $Aug_I(\mathcal{G})$.*

The adjacencies in Lemma 2 capture all dependencies induced by inducing paths, forming the foundation for constructing the skeleton. Then, interventional data help further identify the structures:

**Lemma 3** (When does intervention always alter marginal distribution?). *For any $i, j \in [N]$, $i \in I$, $X_i$ and $X_j$ are adjacent in $\mathcal{M}(Aug_I(\mathcal{G}))$ with a tail at $X_i$, if and only if every inducing path between $X_i$ and $X_j$ begins with a tail in $Aug_{\mathcal{I}}(\mathcal{G})$, i.e., $X_i$ is the ancestor of $X_j$ or is ancestrally selected.*

With Lemma 3, the variant marginal distribution characterizes the ECs of inducing paths starting with a tail. The variability in the conditional distribution is utilized to characterize the ECs as follows:

**Lemma 4** (When does intervention always alter conditional distribution?). *For any $i, j \in [N]$, $i \in I$, $X_i$ and $X_j$ are adjacent in $\mathcal{M}(Aug_I(\mathcal{G}))$ with an arrowhead at $X_i$, if and only if every inducing path between $X_i$ and $X_j$ begins with an arrowhead in $Aug_{\mathcal{I}}(\mathcal{G})$, i.e., $X_i$ is a descendant of $X_j$ or $L$.*

With the foundational graphical criteria that are consistent with the MAG construction ready, the graphical criteria for the $\mathcal{FI}$-Markov equivalence learned from data are as follows:

**Theorem 2** (Graphical criteria for $\mathcal{FI}$-Markov equivalence). *Two augmented DAGs $Aug_{\mathcal{I}}(\mathcal{G}_1) = (\psi \cup X \cup L \cup S, E)$ and $Aug_{\mathcal{I}}(\mathcal{G}_2) = (\psi \cup X \cup L^* \cup S^*, E^*)$ are $\mathcal{FI}$-Markov equivalent for a set of*

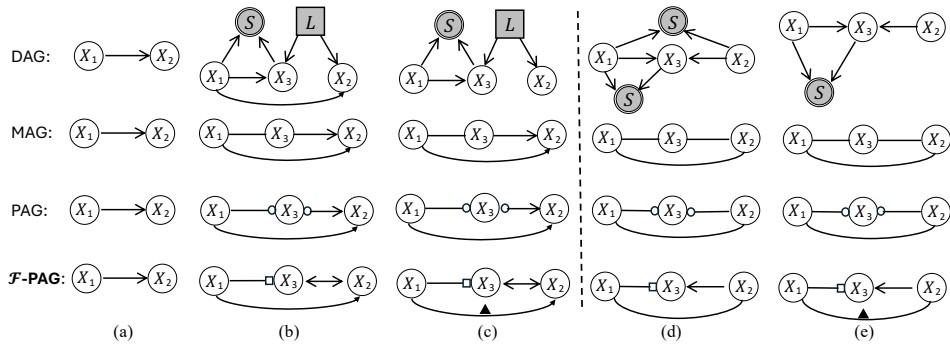

Figure 5: Illustrations of $\mathcal{F}$-PAG graphical representation for the $\mathcal{FI}$-Markov equivalence class.

*targets $\mathcal{I}$ if and only if the corresponding MAGs for $\mathcal{M}_1 = \mathcal{M}(Aug_I(\mathcal{G}_1))$ and $\mathcal{M}_2 = \mathcal{M}(Aug_I(\mathcal{G}_2))$ have the same skeleton and v-structure, and have the same marks and edges among intervened nodes.*

### 3.3.2 $\mathcal{F}$-PAG: Graphical representation for $\mathcal{FI}$-Markov Equivalence

Based on the Markov properties of the augmented DAG in the presence of latent confounders and post-treatment selection, the CI patterns precisely characterize when two such distributions are Markov equivalent in Section 3.3.1. This alignment with our learning objective allows us to recover the EC directly from data. For the graphical representation of the ECs, we follow the conventional approach of using the Partial Ancestral Graph (PAG), defined as follows:

**Definition 4 (Partial ancestral graph).** Let $[\mathcal{M}]$ be the Markov equivalence class of a MAG $\mathcal{M}$. A partial ancestral graph (PAG) for $[\mathcal{M}]$ is a graph $P$ with possibly six kinds of edges: $—, \to, \leftrightarrow$ , $\circ—, \circ—\circ, \circ\to$, such that (1) $P$ has the same adjacencies as $\mathcal{M}$ does; (2) every non-circle mark in $P$ is an invariant mark in $[\mathcal{M}]$. If it is furthermore true that (3) every circle in $P$ corresponds to a variant (indeterminate) mark in $[\mathcal{M}]$, $P$ is called the maximally informative PAG.

Although PAG is a general framework for the graphical representation of DAGs under selection bias and latent confounders, it is designed for ECs, which are maximally informative in CI relations from observational data (v-structure induced independence). This limitation usually results in dependencies induced by broad inducing paths represented by $\circ—\circ$. With the discussed characterizations of Markov properties in Section 3.2, **PAG is too broad to represent $\mathcal{FI}$-Markov equivalence**. For example, in Figure 5(b) and (c), the presence or absence of a causal link between $X_1$ and $X_2$ results in the same PAG. However, Figure 5(c) can be distinguished with interventional data as discussed in Figure 4 (b). To describe the reduced $\mathcal{FI}$-Markov equivalence, we proposed the $\mathcal{F}$-PAG defined as follows:

**Definition 5 ($\mathcal{F}$-Partial Ancestral Graph).** A $\mathcal{F}$-Partial Ancestral Graph, denoted as $\mathcal{G}_p$, is a graphical representation derived from a DAG with latent and selection variables. It captures conditional independence relationships and consists of four types of marks (tail $—$, arrowhead $>$, square $\square$, and circle $\circ$), and eight types of edges ($\overset{\blacktriangle}{\to}, \overset{\blacktriangle}{—}, —, \to, \leftrightarrow, \square—, \square—\square, \square\to, \circ—, \circ—\circ, \circ\to$).

The mark $\square$ denotes a node with at least one tail and at least one arrowhead, $\overset{\blacktriangle}{\to}$ (Figure 5(c)) and $\overset{\blacktriangle}{—}$ (Figure 5(e)) represent inducing paths that have the same CI patterns with $\to, —$, but without a direct causal link and selection separately in between. These two types of inducing paths can be identified only through the graphical conditions involving **Type I** inducing nodes defined as follows:

**Definition 6 (Inducing Node).** In an $\mathcal{F}$-PAG, the nodes are referred to as inducing nodes if and only if the non-endpoint nodes on the inducing path are characterized either by an incoming arrowhead into a square ($\to \square$ **Type I**) or by adjacent two squares ($\square\square$ **Type II**).

For example, in Figure 5(b), non-endpoint node $X_3$ is a **Type I** inducing node. With the advanced graphical representation in place, $\mathcal{FI}$-Markov equivalence can be expressed more clearly, allowing us to distinguish whether the observed dependence arises from genuine causal relations (a) and (b), direct selection (d), or from equivalent inducing paths (c) and (e) as shown in Figure 5.

## 4 Algorithm: $\mathcal{F}$-FCI

In this section, we propose a novel Algorithm 1, named $\mathcal{F}$ine-grain FCI ($\mathcal{F}$-FCI). Using Markov properties of the augmented DAG in Section 3, $\mathcal{F}$-FCI learns causal relations, latent confounders,

---

**Algorithm 1:** $\mathcal{F}$-FCI: Algorithm for learning $\mathcal{F}$-PAG

---

**Input:** Observational and interventional data $\{p^{(k)}\}_{k=0}^{K}$ over $X_{[N]}$ with interventional targets $\mathcal{I}$.
**Output:** A fine-grained partially ancestral graph ($\mathcal{F}$-PAG $\mathcal{G}_p$) over vertices $X$.

**Step 1: Get skeleton from pure observational data.** $\mathcal{G}_p^{(0)} \leftarrow \texttt{FCI}_{ske}(p^{(0)}))$.
**Step 2: Get $\mathcal{F}$-PAG orientation over $X_{\bigcup \mathcal{I}}$ from interventional data. for** $1 \leq i < j \leq K$ **do**

    **Step 2.1: Capture CI patterns between $\psi$ and $X_{\bigcup \mathcal{I}}$.**

    **foreach** *condition set* $C \subseteq \{X_n : X_n \in AllPaths(\mathcal{G}_p^{(0)}, X_{\mathcal{I}(i)}, X_{\mathcal{I}(j)})\} \setminus \{X_{\mathcal{I}(i)}, X_{\mathcal{I}(j)}\}$ **do**

         $CIs = \{ \texttt{CI}\,(\psi_{I(i)}, X_{I(j)} \mid C), \texttt{CI}\,(\psi_{I(i)}, X_{I(j)} \mid X_{I(i)}, C), \texttt{CI}\,(\psi_{I(j)}, X_{I(i)} \mid C),$
         $\texttt{CI}\,(\psi_{I(j)}, X_{I(i)} \mid X_{I(j)}, C)\}$, **If** no more paths can be blocked **then** break;

    **Step 2.2: Orient the skeleton between $X_{I(i)}$ and $X_{I(j)}$.**

        **if** $CIs == (\not\perp\!\!\!\perp, \perp\!\!\!\perp, \perp\!\!\!\perp, \not\perp\!\!\!\perp)$ **then** Orient $X_{I(i)} \rightarrow X_{I(j)}$
        **if** $CIs == (\perp\!\!\!\perp, \not\perp\!\!\!\perp, \perp\!\!\!\perp, \not\perp\!\!\!\perp)$ **then** Orient $X_{I(i)} \leftrightarrow X_{I(j)}$
        **if** $CIs == (\not\perp\!\!\!\perp, \not\perp\!\!\!\perp, \perp\!\!\!\perp, \not\perp\!\!\!\perp)$ **then** Orient $X_{I(i)} \circ\!\!\rightarrow X_{I(j)}$
        **if** $CIs == (\not\perp\!\!\!\perp, \perp\!\!\!\perp, \not\perp\!\!\!\perp, \perp\!\!\!\perp)$ **then** Orient $X_{I(i)} \!-\! X_{I(j)}$
        **if** $CIs == (\not\perp\!\!\!\perp, \perp\!\!\!\perp, \not\perp\!\!\!\perp, \not\perp\!\!\!\perp)$ **then** Orient $X_{I(i)} \!-\!\square X_{I(j)}$
        **if** $CIs == (\not\perp\!\!\!\perp, \not\perp\!\!\!\perp, \not\perp\!\!\!\perp, \not\perp\!\!\!\perp)$ **then** Orient $X_{I(i)} \square\!\!-\!\!\square X_{I(j)}$

    **Step 2.3: Refine the orientation.** Identify causal relations in between for $X_{I(i)} \circ\!\!\rightarrow X_{I(j)}$.

    **foreach** *inducing path between the node pairs with interventional data* $(X_{I(i)}, X_{I(j)})$
    *marked with $\rightarrow$ or $-$ from **Step 2.2*** **do**

        Detect if the path has non-endpoints vertex and **Type I** inducing nodes.
        **If** $\exists$ **Type I** inducing node $X_n$ with $X_{I(i)} \rightarrow X_n \square\!\!-\! X_{I(j)}$ **then** $\texttt{CI}\,(\psi_n, X_{I(i)})$.
            **If** $\psi_n \perp\!\!\!\perp X_{I(i)}$ **then** update $X_{I(i)} \!-\! X_{I(j)}$ to $X_{I(i)} \overset{\blacktriangle}{-} X_{I(j)}$.
        **If** $\exists$ **Type I** inducing node $X_n$ with $X_{I(i)} \!-\!\square X_n \leftrightarrow X_{I(j)}$ **then** $\texttt{CI}\,(\psi_n, X_{I(j)})$.
            **If** $\psi_n \perp\!\!\!\perp X_{I(j)}$ **then** update $X_{I(i)} \rightarrow X_{I(j)}$ to $X_{I(i)} \overset{\blacktriangle}{\rightarrow} X_{I(j)}$.

    **Step 2.4: Update the $\mathcal{F}$-PAG.** $\mathcal{G}_p^{(k)} \leftarrow \mathcal{G}_p^{(k-1)}$
**Step 3: Get $\mathcal{F}$-PAG orientation over $X_{[N]/\bigcup \mathcal{I}}$.** Apply the orientation rules of FCI among
$X_{[N]/\bigcup \mathcal{I}}$ and apply the invariance rules in Theorem 1 (the *see-see*, *do-see*, and *do-do* rules in
(Kocaoglu et al., 2019)) between $X_{\bigcup \mathcal{I}}$ and $X_{[N]/\bigcup \mathcal{I}}$ in $\mathcal{G}_p^{(K)}$. Update $\mathcal{G}_p \leftarrow \mathcal{G}_p^{(K)}$
**return** $\mathcal{G}_p$

---

and post-treatment selection up to the $\mathcal{FI}$-Markov equivalence class, from both observational and interventional data. We assume faithfulness, i.e., no CIs beyond those implied by the graph.

The first step is to recover the undirected skeleton from observational data $p^{(0)}$, since it yields the sparsest graph encoding all inducing paths. We then adopt the standard constraint-based skeleton discovery procedure (e.g., as in FCI) under our general graphical assumptions to obtain this skeleton. Step 2 consists of four sub-steps. **Step 2.1** captures CI patterns reflecting marginalized changes between observational $p^{(0)}$ and interventional data $\{p^{(k)}\}_{k=1}^{K}$. In **Step 2.2**, we then leverage the captured CI patterns to orient edges incident to the intervened variables $X_{\bigcup \mathcal{I}}$, using the orientation rules summarized in Figure 4. In particular, the rule $\circ\!\!\rightarrow$ uses $\circ$ instead of $\square$ because the existence of an inducing path beginning with a tail in between is uncertain when based solely on the marginalized changes between observation and intervention. For example, the structure $\psi_1 \rightarrow X_1 \leftarrow L \rightarrow X_2$. When $X_1$ is under selection, one observes $\psi_1 \not\perp\!\!\!\perp X_2$ regardless of whether $X_1$ is conditioned on. Then, this structure cannot be distinguished from the structure of the latent with causal relation in Figure 4(d), represented by $\circ\!\!\rightarrow$.

To go beyond CI patterns and identify real structure, in **Step 2.3**, $\mathcal{F}$-FCI firstly addresses the uncertainty of $\circ\!\!\rightarrow$ by blocking selection on latent confounders via marginalized changes from two hard interventions. Then, **Type I** inducing nodes along inducing paths between intervened variables are detected to disambiguate cases that CI patterns of endpoints alone cannot distinguish. This procedure is valid for inducing paths containing more than one non-endpoint vertex and including at least one **Type I** inducing node. For example, graph (b) in Figure 4 share the identical CI patterns regardless of whether a direct or indirect causal link exists between $X_1$ and $X_2$. By utilizing changes of hard intervention on the **Type I** inducing node $X_3$, we can test for $\psi_3 \perp\!\!\!\perp X_2 \mid S$ to determine whether a true causal relation exists. Likewise, direct selection (Figure 4(f)) becomes identifiable under the same rationale. Specialized edge marks $\overset{\blacktriangle}{\rightarrow}$ and $\overset{\blacktriangle}{-}$ are established to represent the inducing paths in Figure 5.

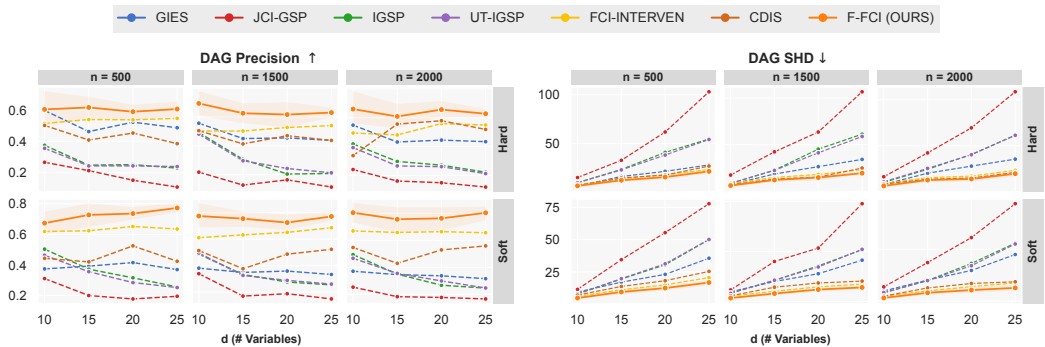

Figure 6: Comparison results in identifying causal relations under DAG Precision and DAG SHD metrics. All values are averaged over 10 graphs. Error bars represent the 95% confidence interval.

By explicitly orienting edges between intervened node pairs, the core contribution of our algorithm, we subsequently apply the standard FCI orientation rules and rules of invariance to all remaining edges: those between intervened and unintervened nodes, as well as those among unintervened nodes in **Step 3**. The extra orientations recovered from interventional data furnish richer structural information than v-structures identified from only observational data.

**Theorem 3** (Soundness of $\mathcal{F}$-FCI). *Let $\mathcal{G}_p$ be the output $\mathcal{F}$-PAG of Algorithm 1 with oracle CI tests on multi-distribution data $\{p^{(k)}\}_{k=0}^K$ given by $(\mathcal{G}, \mathcal{I})$. $\mathcal{G}_p$ is consistent with augmented DAG $Aug_{\mathcal{I}}(\mathcal{G})$ in arrowhead, tails, square, and structures of paths $\overset{\blacktriangle}{\rightarrow}, \overset{\blacktriangle}{-}$ among intervened variables.*

**Theorem 4** (Completeness of $\mathcal{F}$-FCI). *Let $\hat{\mathcal{G}}_p$ be the output of Algorithm 1 with oracle CI tests on multi-distribution data $\{p^{(k)}\}_{k=0}^K$ given by $(\mathcal{G}, \mathcal{I})$. Each type of substructures represented by tail, arrowhead, square, $\overset{\blacktriangle}{\rightarrow}, and \overset{\blacktriangle}{-}$ between a pair of intervened nodes in the corresponding augmented DAG of $\hat{\mathcal{G}}_p$ can be identified by different types of CI patterns.*

## 5 EXPERIMENTS

In this section, we present empirical studies on simulations and real-world data to demonstrate that $\mathcal{F}$-FCI identifies causal relations in the presence of latent confounders and post-treatment selection.

### 5.1 SIMULATIONS

We conduct simulations to validate the effectiveness of our proposed $\mathcal{F}$-FCI. We compare $\mathcal{F}$-FCI against strong baselines in interventional causal discovery, including GIES (Hauser & Bühlmann, 2012), IGSP (Wang et al., 2017), UT-IGSP (Squires et al., 2020), JCI-GSP (Mooij et al., 2020), FCI with interventional data (FCI-interven) (Kocaoglu et al., 2019), and CDIS (Dai et al., 2025).

Following the data-generating procedure in Definition 1, we begin by randomly sampling Erdös–Rényi graphs with an average degree of 2 as the ground truth DAG for $\{X_i\}_{i=1}^N$. We then randomly generate 2 or 3 selection variables, each with two randomly chosen parents from $\{X_i\}_{i=1}^N$, and 2 or 3 latent confounders, each with two randomly chosen children. Finally, we simulate general SEMs $X_i = f(\hat{X}_{pa_{\mathcal{G}}(i)}) + \epsilon_i$, $\epsilon_i$ is sampled from $Unif([0,2] \cup [2,4])$, and select samples with $\sum f_s(X_i)$ that fall within a predefined interval, where $f$ and $f_s$ are randomly drawn from linear, square, sin and tanh.

To evaluate the effectiveness of $\mathcal{F}$-FCI [1] in identifying causal relations despite the presence of latent confounders and post-treatment selection, we report the main Precision and Structural Hamming Distance (SHD) metrics compared with baseline methods in Figure 6 (the F1-score and recall are given in Figure 10 in Appendix D). The experimental results demonstrate that $\mathcal{F}$-FCI outperforms baselines with an average precision of over 5% in most configurations and lower SHD. These observations validate the effectiveness of $\mathcal{F}$-FCI in identifying true causal relations, whereas baselines may infer spurious ones induced by latent confounders and post-treatment selection. Moreover, the **robustness of $\mathcal{F}$-FCI under different noise levels** is evaluated in Figure 12, the **scalability** is evaluated in Figure 11, and **its ability to distinguish post-treatment selection** is assessed in Table 1.

---

[1] A Python implementation of $\mathcal{F}$-FCI is available at `https://github.com/GongxuLuo/F-FCI`.

## 5.2 REAL-WORLD APPLICATIONS

We evaluate the gene regulatory networks (GRNs) of genes using large-scale single-cell gene perturbation data collected from Human Lung Epithelial Cells (HLEC), i.e., Norman datasets (Norman et al., 2019). We report both the regulatory (causal) links and the spurious dependencies induced by post-treatment selection, as identified by $\mathcal{F}$-FCI in Figure 13, and detailed analysis can be found in Appendix D.3. Experimental results are evaluated using prior knowledge provided by Enrichr, a tool that compiles extensive libraries from enrichment experiments (Kuleshov et al., 2016; Xie et al., 2021).

## 6 CONCLUSION AND LIMITATIONS

We introduce a fundamental yet underexplored challenge for causal discovery: post-treatment selection, particularly the often-overlooked quality control constraint that shapes dependencies. We show why existing models fail to handle such bias, propose a new formulation to model post-treatment selection, establish criteria for a novel fine-grained interventional Markov equivalence, and define a corresponding graphical representation. Building on this formulation, we develop a sound and complete algorithm, named $\mathcal{F}$-FCI, that uncovers causal relations and post-treatment selection. Empirical analyses on synthetic and large-scale real-world datasets demonstrate the effectiveness of $\mathcal{F}$-FCI in accurate and robust causal discovery.

The identification of direct causal links and selection structures depends critically on the presence of **Type I** inducing nodes. One future direction is how to identify the causal structure along inducing paths composed solely of **Type II** inducing nodes. In addition, as discussed in (Luo et al., 2025), biological constraints filter out cells and introduce extra dependencies; another extension can be how to distinguish biological constraints from post-treatment selection.

## ACKNOWLEDGEMENTS

We would also like to acknowledge the support from NSF Award No. 2229881, AI Institute for Societal Decision Making (AI-SDM), the National Institutes of Health (NIH) under Contract R01HL159805, and grants from Quris AI, Florin Court Capital, MBZUAI-WIS Joint Program, and the Al Deira Causal Education project.

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

# A    DEFINITIONS

Over the past decades, comprehensive causal discovery frameworks have been established. We now introduce the basic concepts and fundamental definitions for the general setting involving latent confounders and selection bias. To represent the general graph with latent factors, selection bias, and loops, the **mixed** graph is first designed with direct $\rightarrow$, undirected —, and bidirected $\leftrightarrow$ edges for all structural representations. Given a mixed graph $\mathcal{G}_M$ and two adjacent vertices $X_i, X_j \in X$. $X_i$ is a parent of $X_j$ and $X_j$ is a child of $X_i$, if $X_i \rightarrow X_j$. $X_i$ is called a spouse of $X_j$ if $X_i \leftrightarrow X_j$ in $\mathcal{G}_M$. A direct path from $X_1$ to $X_n$ is a sequence of vertices where each $1 \leq i \leq n-1$, $X_i$ is the parent of $X_{i+1}$. $X_i$ is called an ancestor of $X_j$ and $X_j$ is the descendant of $X_i$ if $X_i = X_j$ or there is a directed path from $X_i$ to $X_j$. Let $Ans_{\mathcal{G}_M}(X_i)$ denote the set of ancestors of $X_i$ in $\mathcal{G}_M$. A directed cycle occurs in $\mathcal{G}_M$ when $X_i \rightarrow X_j$ is in $\mathcal{G}_M$ and $X_j \in Ans_{\mathcal{G}_M}(X_i)$. An almost directed cycle occurs when $X_i \leftrightarrow X_j$ is in $\mathcal{G}_M$ and $X_j \in Ans_{\mathcal{G}_M}(X_i)$. The ancestral graph is defined in Definition 8 without considering cycles.

**Definition 7 (d-separation).** If every path from a node in $X$ to a node in $Y$ is d-separated by $Z$, then $X$ and $Y$ are always conditionally independent given $Z$.

**Definition 8 (Ancestral graph** (Zhang, 2008b)**).** A mixed graph is ancestral if and only if the following conditions hold: (1) there is no directed cycle; (2) there is no almost directed cycle; (3) for any undirected edge $X_i—X_j$, $X_i$ and $X_j$ have no parents or spouses.

The other detailed description of the definitions that characterize the graphical causal model is introduced as follows:

**Definition 9 (m-separation).** In a mixed graph, a path $p$ between disjoint subsets of vertices $X_A$ and $X_B$ is active (m-connecting) relative to a set of vertices $X_Z$ ($X_A, X_B \notin X_Z$) if and only if (1) every non-collider on $p$ is not in $X_Z$; (2) every collider on $p$ has a descendant in $X_Z$. $X_A$ and $X_B$ are said to be m-separated by $X_Z$ if and only if there is no active path between them relative to $X_Z$.

**Definition 10 (Maximality).** An ancestral graph is said to be maximal if any two non-adjacent vertices, there is a set of vertices that m-separates them.

**Definition 11 (Markov equivalence).** Two MAGs $\mathcal{M}_1$, $\mathcal{M}_2$ are Markov Equivalent if for any three disjoint sets of vertices $X_A$, $X_B$, $X_C$, $X_A$ and $X_B$ are m-separated by $X_C$ in $\mathcal{M}_1$ if and only if they are m-separated by $X_C$ in $\mathcal{M}_2$ as well.

**Definition 12 (Augmented DAG).** For a DAG $\mathcal{G}$ over $X \cup L \cup S$ and intervention target $I \subset [N]$ the *augmented graph* $Aug_{\mathcal{I}}(\mathcal{G})$ is a DAG with vertices $\psi \cup X \cup L \cup S \cup \epsilon \cup \xi$, where:

- $\psi = \{\psi_{I^{(k)}}\}_{k=1}^K$ is a set of exogenous binary indicators for the representation of marginal changes between two environments (observation-intervention or intervention-intervention, two hard interventions are a special case), pointing to the corresponding intervened variable $X_{I^{(k)}}$;

- $X = (X_i)_{i=1}^N$ are the observed variables, pure observational or interventional;

- $L = \{L_1, L_2, ..., L_R\}$ and $S = \{S_1, S_2, ..., S_T\}$ represent the sets of latent confounders and post-treatment selection respectively in the hidden world that influence the observations;

- $\epsilon$ denotes the exogenous noise term for variables, whether observed or hidden.

With the defined variables, $Aug_{\mathcal{I}}(\mathcal{G})$ consists of the following edges:

- Direct causal effect edges: for each $X_i \rightarrow X_j \in \mathcal{G}$ with $i, j \in [N]$, add $X_i \rightarrow X_j \in \mathcal{G}^{Aug}$, for each $X_i \in \mathcal{G}$ with $i \in [N]$ is directly influenced by $L_r$, $r \in [R]$, add $L_r \rightarrow X_i$;

- Selection edge: for each $X_i$ with $i \in [N]$ is directly selected by $S_t$, $t \in [T]$, add $X_i \rightarrow S_t$;

- Edges representing common exogenous influences: $\{\epsilon_i \rightarrow X_i\}_{i \in [N]}$;

- Edges representing mechanism changes due to the intervention: $\{\psi_i \rightarrow X_i\}_{i \in I}$.

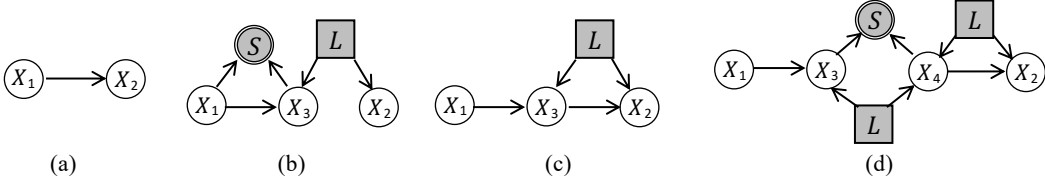

Figure 7: Illustrative examples of inducing paths with $X_1$ is an ancestor of $X_2$ in subfigures(a)&(c) or is ancestrally selected, shown in subfigures (b)&(d).

## B PROOFS OF MAIN RESULTS

Since the technical development of this paper is fundamentally built upon the characterization of the augmented DAG, we focus on proving only those results that do not directly follow from the basic properties of augmented DAGs. Specifically, the results stated in Theorem 1, Lemma 1, and Lemma 2 are immediate results of the definition of augmented DAGs (Spirtes et al., 2000; Korb et al., 2004; Zhang, 2008b; Kocaoglu et al., 2019). Therefore, their proofs are omitted.

**Lemma 3** (When does intervention always alter marginal distribution?). *For any $i, j \in [N]$, $i \in I$, $X_i$ and $X_j$ are adjacent in $\mathcal{M}(Aug_I(\mathcal{G}))$ with a tail at $X_i$, if and only if every inducing path between $X_i$ and $X_j$ begins with a tail in $Aug_\mathcal{I}(\mathcal{G})$, i.e., $X_i$ is the ancestor of $X_j$ or is ancestrally selected.*

*Proof of Lemma 3.* Before proceeding, we note a special case where the marginal distribution changes: when $i \in I$ with $\psi_i \to X_i$. In this case, it is straightforward that $p^{(0)}(X_i) \neq p^{(k)}(X_i), I^{(k)} = \{i\}$, as $X_i$ is intervened upon. We omit the proof as it is self-evident.

1. **('⇐' direction)** For any $i, j \in [N]$, the paths between $X_i$ and $X_j$ are inducing paths that start with tails directed outward from $X_i$ toward $X_j$ in the augmented DAG $Aug_\mathcal{I}(\mathcal{G})$. Because the inducing path starts with a tail and its definition that every collider is the ancestor of either $X_i$, $X_j$, or $S$, this implies that $X_i$ is either an ancestor of $X_j$ or directly selected, as illustrated in Figure 7. Otherwise, the path will be blocked, conflicting with the definition of the inducing path. Furthermore, only the ancestral relationship and selection propagate distribution changes. Therefore, when $X_i$ is intervened upon, distribution changes propagate along the inducing paths via the outgoing tails from $X_i$, thereby altering the marginal distribution of $X_j$. Whin in the augmented DAG $Aug_\mathcal{I}(\mathcal{G})$, $\psi_i \to X_i$, as all inducing paths orient outward from $X_i$ with tails, $\psi_i \not\perp\!\!\!\perp X_j$. Then, $p^{(0)}(X_j) \neq p^{(k)}(X_j), I^{(k)} = \{i\}$.

2. **('⇒' direction)** By contradiction, suppose $X_i \notin \{Ans_{Aug_\mathcal{I}(\mathcal{G})}(X_j) \cup Ans_{Aug_\mathcal{I}(\mathcal{G})}(S)\}$, as the paths between $X_i$ and $X_j$ are inducing path, compromising by inducing nodes, that can not be blocked, which requires both collider and tails occurs.
   - When the arrowhead emerges outward from $X_i \to$, the formation of colliders requires another arrowhead at the same intermediate node from $\leftarrow\!\circ X_j$. Then, according to the definition of the inducing path, whether the tail of the middle node from causation or selection, it violates the condition $X_i \notin \{Ans_{Aug_\mathcal{I}(\mathcal{G})}(X_j) \cup Ans_{Aug_\mathcal{I}(\mathcal{G})}(S)\}$.
   - When the collider formation is from $X_i \leftrightarrow$ and $\leftarrow X_j$, the tail between the intermediate node and $X_j$ only origin from selection. When the collider formation is from $X_1 \leftrightarrow$ and $\leftrightarrow X_j$, the edge with tails at the collider toward $X_j$ can arise from both causation and selection. Then, the inducing paths starting from $X_i$ with the one inducing node are represented by $X_i \leftrightarrow \square \to X_j$ and $X_i \leftrightarrow \square\!-\!\square X_j$ separately. However, in both structures, $p^{(0)}(X_j) = p^{(k)}(X_j), I^{(k)} = \{i\}$, which contradicts with the changes in marginal distribution.

□

**Lemma 4** (When does intervention always alter conditional distribution?). *For any $i, j \in [N]$, $i \in I$, $X_i$ and $X_j$ are adjacent in $\mathcal{M}(Aug_I(\mathcal{G}))$ with an arrowhead at $X_i$, if and only if every inducing path between $X_i$ and $X_j$ begins with an arrowhead in $Aug_\mathcal{I}(\mathcal{G})$, i.e., $X_i$ is a descendant of $X_j$ or L.*

*Proof of Lemma 4.*

1. **('⇐' direction)** For any $i, j \in [N]$, $i \in I$, the paths between $X_i$ and $X_j$ are inducing paths in the augmented DAG $Aug_\mathcal{I}(\mathcal{G})$. If $X_i$ is the descendant of $X_j$ or L, there must be an

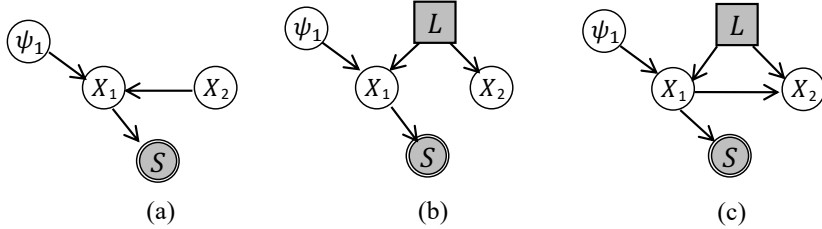

(a)                            (b)                            (c)

Figure 8: Cases of when selection introduce Algorithm 1. (a) is not distorted by Y structure $\psi_i \rightarrow X_i \leftarrow X_j, X_i \rightarrow S$, due the causation between $X_1$ and $X_2$. However, a direct causation can not be identified in (b) and (c).

arrowhead pointing toward $X_i$. Then, with inducing paths containing one middle node as examples, the path may take the form $X_i \leftarrow \square \leftarrow \circ X_j$ or $X_i \leftrightarrow \square\!-\!\circ X_j$ . When $X_i$ is intervened upon, the changes in p($X_i$) cannot propagate along the inducing path as the arrowhead pointing to $X_i$ blocks the propagation, resulting in the changes of the conditional distribution p($X_j \mid X_i$). Accordingly, in the augmented DAG $Aug_{\mathcal{I}}(\mathcal{G})$, we have $\psi_i \not\perp\!\!\!\perp X_j \mid X_i$, and thus $p^{(0)}(X_j \mid X_i) \neq p^{(k)}(X_j \mid X_i), I^{(k)} = \{i\}$.

2. (**'$\Rightarrow$' direction**) By contradiction, suppose $X_i \notin \{Des_{Aug_{\mathcal{I}}(\mathcal{G})}(X_j) \cup Des_{Aug_{\mathcal{I}}(\mathcal{G})}(L)\}$. To satisfy the requirements for inducing paths between $X_i$ and $X_j$, every non-endpoint node must be a collider with a tail oriented outward to $X_j$ or $S$. Consequently, the arrowhead only can arise from $X_i \rightarrow$, yielding an inducing path of the form $X_i \rightarrow \square\!-\!\circ X_j$. In this case, however, $p^{(0)}(X_j \mid X_i) = p^{(k)}(X_j \mid X_i), I^{(k)} = \{i\}$, which conflicts with the requirement that conditional distribution changes.

$\square$

**Theorem 2** (Graphical criteria for $\mathcal{FI}$-Markov equivalence). *Two augmented DAGs $Aug_{\mathcal{I}}(\mathcal{G}_1) = (\psi \cup X \cup L \cup S,\ E)$ and $Aug_{\mathcal{I}}(\mathcal{G}_2) = (\psi \cup X \cup L^* \cup S^*,\ E^*)$ are $\mathcal{FI}$-Markov equivalent for a set of targets $\mathcal{I}$ if and only if the corresponding MAGs for $\mathcal{M}_1 = \mathcal{M}(Aug_I(\mathcal{G}_1))$ and $\mathcal{M}_2 = \mathcal{M}(Aug_I(\mathcal{G}_2))$ have the same skeleton and v-structure, and have the same marks and edges among intervened nodes.*

*Proof of Theorem 2.* If two augmented DAGs $Aug_I(\mathcal{G}_1)$ and $Aug_I(\mathcal{G}_2)$ are $\mathcal{FI}$-Markov equivalent, they have the same CI patterns for the paths between any pair of nodes $X_i$ and $X_j$, where $i, j \in I$. In total, there are three types of node marks: tail, arrowhead, and both; these can be uniquely learned from data based on the Lemmas 2 to 4.

- **Arrowhead**. If the arrowhead points into $X_i$, no matter from causation or latent confounders, this implies that $X_i$ cannot be the ancestor of $X_j$ or the ancestor of $S$ on the inducing paths. Therefore, following the construction rules of MAG (Zhang, 2008b), it must be oriented as $X_i \leftarrow$ in MAG representation.
- **Tail.** If tails are oriented outward from $X_i$, as all the paths between $X_i$ and $X_j$ must be the inducing path, when the tail arises from $X_i \rightarrow$, the formation of an inducing node like $X_3$ in Figure 7(d) needs at least one tail and one arrowhead. When the tail originates from selection, then $X_i \in Ans_{Aug_{\mathcal{I}}(\mathcal{G})}(S)$, resulting in a tail pointing out $X_i$ in MAG representation. When the tail comes from causation, then $X_i \in Ans_{Aug_{\mathcal{I}}(\mathcal{G})}(X_j)$. In either case, the orientation is consistent with MAG construction rules for tails.
- **Both arrowhead and tail**. When both an arrowhead points to $X_i$ and a tail points outward from $X_i$, only two types of combinations are possible: selection with arrowhead, or causation with latent confounders. In both cases, the MAG representation assigns a tail at $X_i$. This is because, according to the formation of the inducing node, causation and latent confounders provide only an arrowhead. Regardless of thether the tail from causation or selection, $X_i \in \{Ans_{Aug_{\mathcal{I}}(\mathcal{G})}(X_j) \cup Ans_{Aug_{\mathcal{I}}(\mathcal{G})}(S)\}$. Then, both are presented by tail in MAG representation.

The consistency in the MAG construction rules of augmented DAGs ensures that $M_1$ and $M_2$ have the same skeleton and v-structure in their MAG presentations, as well as the same marks for intervened nodes. $\square$

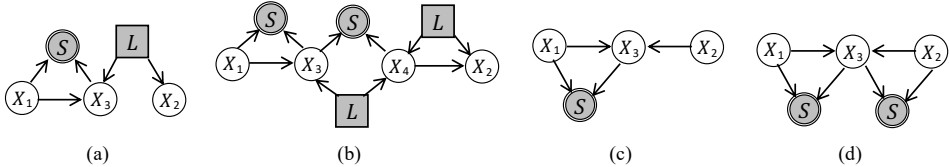

Figure 9: Inducing path between $X_1$ and $X_2$ with **Type I** inducing node (a,c) or not (b,d).

**Theorem 3** (Soundness of $\mathcal{F}$-FCI). *Let $\mathcal{G}_p$ be the output $\mathcal{F}$-PAG of Algorithm 1 with oracle CI tests on multi-distribution data $\{p^{(k)}\}_{k=0}^K$ given by $(\mathcal{G}, \mathcal{I})$. $\mathcal{G}_p$ is consistent with augmented DAG $Aug_{\mathcal{I}}(\mathcal{G})$ in arrowhead, tails, square, and structures of paths $\overset{\blacktriangle}{\rightarrow}, \overset{\blacktriangle}{-}$ among intervened variables.*

The soundness of Algorithm 1 indicates that the output arrowhead, tail, square, and $\overset{\blacktriangle}{\rightarrow}, \overset{\blacktriangle}{-}$ are consistent with the corresponding structures in the augmented DAG.

*Proof of Theorem 3.*

- **Tail.** The $\mathcal{F}$-PAG is learned according to the CI results that align with d-separation in the augmented DAG $Aug_I(\mathcal{G})$ as analyzed in Figure 4 under the faithfulness assumption. With the theoretical guarantee proved in Lemma 3 and Lemma 4, changes in marginal distribution $p^{(0)}(X_j) \neq p^{(k)}(X_j), I^{(k)} = \{i\}$ and invariance in conditional distribution $p^{(0)}(X_j \mid X_i) = p^{(k)}(X_j \mid X_i), I^{(k)} = \{i\}$ appear, if and only if the all inducing path between $X_i$ and $X_j$ start with tail point outward from $X_i$, resulting in $\psi_i \not\perp\!\!\!\perp X_j$ and $\psi_i \perp\!\!\!\perp X_j \mid X_i$. The CIs can uniquely identify the tail; others will exhibit different CIs.

- **Arrowhead.** Similarly, the unique invariant marginal distribution $p^{(0)}(X_j) = p^{(k)}(X_j), I^{(k)} = \{i\}$ and variant conditional distribution $p^{(0)}(X_j \mid X_i) \neq p^{(k)}(X_j \mid X_i), I^{(k)} = \{i\}$ only from inducing paths start with the arrowhead in the augmented DAG.

- **Square.** When both tails and arrowheads appear, the node is marked with a square, indicating variation in both marginal distribution $p^{(0)}(X_j) \neq p^{(k)}(X_j), I^{(k)} = \{i\}$ and conditional distribution $p^{(0)}(X_j \mid X_i) \neq p^{(k)}(X_j \mid X_i), I^{(k)} = \{i\}$. This allows the square mark to be uniquely identified. However, there is a special case that needs further discussion. In the augmented DAG $Aug_I(\mathcal{G})$ with $\psi_1 \rightarrow X_1$, if $X_1 \rightarrow S$ and $X_1 \leftarrow$, they form the Y-structure at $X_1$ such as Figure 8(a). Here, as the existence dependence induced by $X_2 \rightarrow X_1$, the extra dependence induced by selection in the Y-structure does not distort the final results, so the resulting edge remains $\square\!\!-\!$. Similarly, the extra dependence induced by Y-structure does not affect the identifiability of $\square\!\!-\!\!\square$ as well, as the existence of the inducing paths starting from both tail and arrowhead in between. However, when the arrowhead into $X_1 \leftarrow$ arises from latent confounders as shown in Figure 8(b,c), the causal relation becomes non-identifiable regardless of whether $X_1 \rightarrow X_2$ holds. In this case, selection contributes a tail and makes the path unblocked; the d-connection between $L$ and $\psi_i$ results in spurious causation, but no inducing path begins with a tail between $X_1$ and $X_2$. This is because of the effect of selection on $L$. Fortunately, we found that hard intervention can block the selection on latent confounders. The marginalized changes from different hard interventions help us identify if there exis the causal link in between. Therefore, the soundness is up to the square of $\square\!\!-\!$ and $\square\!\!-\!\!\square$.

- $\overset{\blacktriangle}{\rightarrow}$ . The identification of $\overset{\blacktriangle}{\rightarrow}$ needs every inducing path to have at least one **Type I** inducing node with corresponding interventional data available, such as $X_3$ of Figure 9(a). If all the nodes are **Type II** inducing nodes as shown in (b), although interventional data is available, both the marginal distribution p($X_1$) and p($X_2$) change no matter $X_1 \rightarrow X_2$ or not, contradicting the identification $\overset{\blacktriangle}{\rightarrow}$.

- $\overset{\blacktriangle}{-}$. As the identification rule condition of $\overset{\blacktriangle}{-}$ is the same as $\overset{\blacktriangle}{\rightarrow}$ . If there is a **Type I** inducing node like $X_3$ of Figure 9(c), the existence of direct selection can be identified by intervening on $X_3$. If there is only a **Type II** inducing node like $X_3$ in (d), the change of p($X_2 \mid do(X_3)$) can be due to the selection between $X_2$ and $X_3$, and cannot identify the existence of direct selection of $X_1 \rightarrow S \leftarrow X_2$.

$\square$

**Theorem 4** (Completeness of $\mathcal{F}$-FCI). *Let $\hat{\mathcal{G}}_p$ be the output of Algorithm 1 with oracle CI tests on multi-distribution data $\{p^{(k)}\}_{k=0}^K$ given by $(\mathcal{G}, \mathcal{I})$. Each type of substructures represented by tail, arrowhead, square, $\overset{\blacktriangle}{\rightarrow}, and \overset{\blacktriangle}{-}$ between a pair of intervened nodes in the corresponding augmented DAG of $\hat{\mathcal{G}}_p$ can be identified by different types of CI patterns.*

*Proof of Theorem 4.* To establish the completeness, we need to show that the $\mathcal{F}$-PAG returned by $\mathcal{F}$-FCI is maximally informative when interventional data is available. Based on the proof of Lemmas 2 to 4, the identifiability of learning the fundamental node marks, i.e., arrowhead, tail, and square, from data is guaranteed. Therefore, among the intervened variables, all the node marks can be identified. Moreover, with the identification guarantee of node markers, the identification of $\overset{\blacktriangle}{\rightarrow}$ and $\overset{\blacktriangle}{-}$ can be guaranteed as well, this is because the condition of **Type I** is detectable based on the identification of node marks. For the node marks without interventional data, the completeness was established with arrowheads in (Ali et al., 2012) and tails in (Zhang, 2008b). For the node marks for node pairs where one of them is intervened, a.k.a. do-see rules, the completeness was established in (Kocaoglu et al., 2019). □

## C  RELATED WORK

In this section, we provide a comprehensive review of the relevant literature.

**When only pure observational data is available.** There are constraint-based causal discovery algorithms (Spirtes et al., 2001; Luo et al., 2025; Li et al., 2024b), score-based algorithms (Chickering, 2002; Li et al., 2024a; Ziu et al., 2024), and methods that utilize properties of functional forms in the underlying causal process (Shimizu et al., 2006; Hoyer et al., 2008; Zhang & Hyvarinen, 2012). The corresponding Markov equivalence characterization can be referred to (Verma & Pearl, 1991; Meek, 1995; Andersson et al., 1997; Robins et al., 2000; Friedman et al., 2000; Brown et al., 2005).

**Two kinds of interventions.** When interventional data is available, two types of interventions have been proposed to model data generation: *hard* (or *perfect*) interventions and *soft* (or *imperfect*) interventions, also referred to as *mechanism changes*. Hard interventions disrupt the dependence between a target variable and its direct causes, either by *deterministically* fixing its value or by *stochastically* assigning values drawn from an independent distribution (Pearl, 2009; Korb et al., 2004). In contrast, soft interventions preserve the dependence but alter the functional form governing the target variable's causal mechanism (Tian & Pearl, 2001; Eberhardt & Scheines, 2007).

**Early Attempts in Interventional Causal Discovery.** The earliest Bayesian approaches (Cooper & Yoo, 1999; Eaton & Murphy, 2007) estimated the posterior distribution of DAGs using both observational and interventional data. However, these methods did not address key challenges such as identifiability or equivalence class characterization. Tian & Pearl (2001) was the first to explore identifiability and Markov equivalence in interventional causal discovery. They focused on single-variable interventions with mechanism changes (soft interventions) and provided a graphical criterion for determining when two DAGs are indistinguishable, though no formal representation of the resulting equivalence class was introduced.

**Advancements in Hard Interventions.** Hauser & Bühlmann (2012) first characterized Markov equivalence classes (MECs) under hard stochastic, multiple-variable interventions. Their graphical criterion, based on mutilated DAGs (as introduced in Section 2), provides an equivalence class representation using $\mathcal{I}$-essential graphs. This criterion aligns with Tian & Pearl (2001), though the latter focuses solely on single-variable interventions. The proposed GIES algorithm (Hauser & Bühlmann, 2015) integrates conditional independence (CI) relations from different experimental settings. Following this paradigm, related methods have been developed (Tillman & Spirtes, 2011; Claassen & Heskes, 2010). However, Wang et al. (2017) identified consistency issues in GIES when faithfulness assumptions are violated and introduced permutation-based algorithms as an alternative solution.

**When latent confounders are involved.** In the pure observational data and nonparametric causal discovery setting, the frameworks of MAG and FCI have been well established (Richardson &

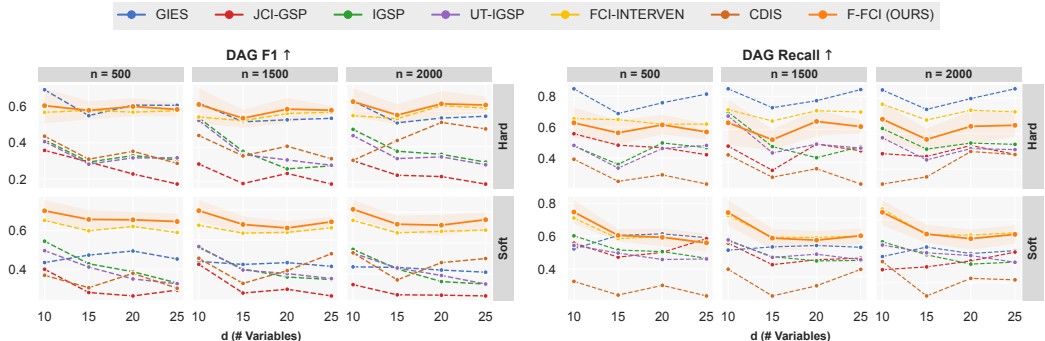

Figure 10: Comparison results in identifying causal relations under DAG F1 score and DAG Recall metrics. All values are averaged over 10 graphs. Error bars represent the 95% confidence interval.

Spirtes, 2002; Zhang, 2008b). For interventional causal discovery, various methods have been proposed to address latent variables based on measuring overlapping variables across different interventions (Hyttinen et al., 2013; Triantafillou & Tsamardinos, 2015; Cao et al., 2024) and invariance (Kocaoglu et al., 2017; Eaton & Murphy, 2007; Magliacane et al., 2016). They are either lying under the umbrellas of FCI and the augmented DAG frameworks or using parametric assumptions.

**When selection is involved** In purely observational and nonparametric causal discovery, selection is typically constrained by structural limitations (Zhang et al., 2016). However, in the context of interventional causal discovery, various methods have been developed to explicitly address selection bias. These methods leverage interventional data to disentangle the effects of selection mechanisms from genuine causal relations Li et al. (2023); Mooij et al. (2020). Similarly, these methods are still limited to equivalence classes under the umbrella of FCI. Moreover, Dai et al. (2025) discussed how selection interacts with intervention, and built a twin interventional graph to model the selection that happens before intervention.

**When both latent confounders and selection are involved.** The presence of latent confounders and selection bias introduces spurious dependencies among observed variables, undermining key assumptions in causal discovery and leading to a loss of identifiability. In the pure observational data, the Fast Causal Inference (FCI) algorithm (Spirtes et al., 2001; Zhang, 2008b) was an early attempt to infer ancestral relationships, but it is constrained to identifying an equivalence class limited by the structural information of v-structures. Moreover, ambiguities remain in handling selection effects. Other approaches have similarly characterized ancestral equivalence classes based on graphical properties, but without fully addressing these challenges Rohekar et al. (2021).

## D SUPPLEMENTARY EXPERIMENTAL DETAILS AND RESULTS

### D.1 SIMULATION

We use the implementation of IGSP, UT-IGSP, and JCI-GSP from the `causaldag` package (Chandler Squires, 2018), and the implementation of CDIS from `https://github.com/MarkDana/CDIS`. We use the Kernel-based CI test (Zhang et al., 2012) to examine nonlinear conditional relations for both baselines and $\mathcal{F}$-FCI. The significance level is set to 0.05.

To comprehensively evaluate the effectiveness of $\mathcal{F}$-FCI in identifying causal relations compared with baselines, besides the experimental result on simulation shown in Figure 6, experimental results on F1-score and recall are shown in Figure 10. All experimental results show that $\mathcal{F}$-FCI can identify the spurious dependence induced by post-treatment selection or by inducing paths illustrated in the motivation examples Figure 1, while baselines cannot.

The simulation results on graphs with 10–25 nodes in the non-parametric setting evaluate the scalability of $\mathcal{F}$-FCI to moderately large graphs. We further conduct experiments with 50 variables, of which 30 have interventional data, while 4–7 selection variables and latent confounders are randomly generated. The resulting $\mathcal{F}$-PAG is shown in Figure 11. In addition, experiments on over 5000 genes further demonstrate the scalability of $\mathcal{F}$-FCI to high-dimensional biological data.

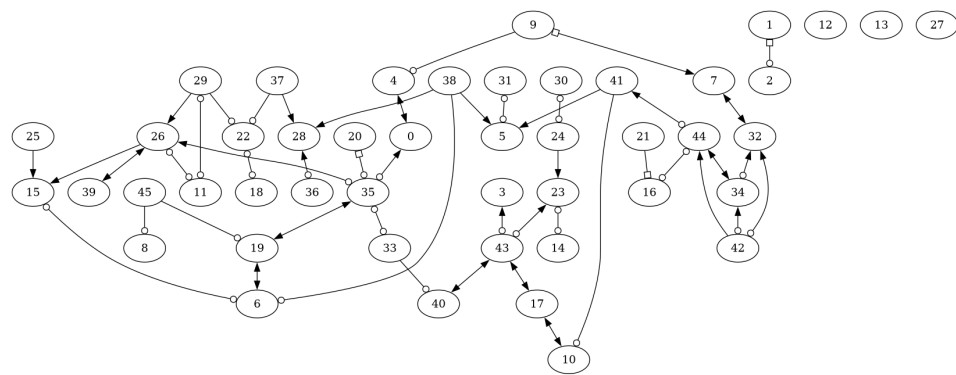

Figure 11: The output $\mathcal{F}$-PAG generated by $\mathcal{F}$-FCI on 50 variables.

Theoretically, the complexity of constraint-based causal discovery methods, including $\mathcal{F}$-FCI, scales with the graph size. However, because the CI tests are independent, the computational burden can be substantially reduced through parallelization. In particular, Fast FCI (Ramsey et al., 2017) can handle millions of variables, and we leverage this implementation for large-scale gene expression data. Based on the previous analysis, these results support the scalability of $\mathcal{F}$-FCI and its practicality for real-world applications.

Table 1: Accuracy % of $\mathcal{F}$-FCI in identifying post-treatment selection on synthetic data. We report the mean and variance values of accuracy across 10 independent graphs for each configuration.

| $n$ \ $|\mathcal{X}|$ | 10 | 15 | 20 | 25 | 10 | 15 | 20 | 25 |
|---|---|---|---|---|---|---|---|---|
| | \multicolumn{4}{c}{**Hard intervention**} | | | | **Soft intervention** | | |
| 500 | 66.7±15.0 | 64.8±9.1 | 75.9±10.5 | 71.7±15.6 | 60.4±24.0 | 57.3±17.3 | 66.7±15.0 | 57.2±17.0 |
| 1,000 | 86.5±4.4 | 72.5±9.1 | 73.1±2.9 | 70.2±10.1 | 63.0±22.7 | 56.7±17.3 | 74.7±8.8 | 52.1±17.5 |
| 1,500 | 93.8±0.9 | 80.9±9.5 | 69.0±7.3 | 70.6±9.6 | 70.0±21.0 | 60.8±16.3 | 75.0±9.6 | 63.9±15.2 |
| 2,000 | 91.6±2.3 | 85.0±1.8 | 75.6±6.8 | 71.3±5.1 | 80.0±16.0 | 63.9±13.1 | 76.3±10.1 | 76.3±8.9 |

Moreover, to verify the effectiveness of $\mathcal{F}$-FCI in handling the post-treatment selection, we report in Table 1 the accuracy of identifying post-treatment selection on simulated data across different configurations. Experimental results demonstrate that under each setting with 2 or 3 selection variables, the accuracy of $\mathcal{F}$-FCI increases with sample size. In most configurations with more than 1000 samples, the accuracy exceeds 70%.

The performance of constraint-based causal discovery methods relies on the accuracy of the conditional independence test. To verify the robustness of $\mathcal{F}$-FCI, we conduct experiments across different noise levels of $X$, where the parameters of the uniform distribution are randomly sampled from the following settings, i.e., $X \sim U([0,1] \cup [1,2])$, $X \sim U([0,2] \cup [2,4])$, and $X \sim U([0,3] \cup [3,6])$. Experimental results on DAG Precision, DAG $F_1$ score, DAG Recall, and DAG SHD are reported in Figure 12. Experimental results show that $\mathcal{F}$-FCI is stable in precision across different noise levels. The low noise level (green line) results in a relatively higher recall. This is because when the noise level is low, as in $X \sim U([0,1] \cup [1,2])$, the variance is smaller than in other settings. Consequently, dependence patterns are more easily detected, leading to the recovery of more edges and thereby yielding higher recall.

## D.2 EXPANDED EVALUATION AND ROBUSTNESS ANALYSES OF $\mathcal{F}$-FCI

To further assess the ability of $\mathcal{F}$-FCI to handle post-treatment selection, we compared $\mathcal{F}$-FCI with FCI-INTERVEN that can only handle latent confounders but not post-treatment selection while increasing the number of selection variables. Experimental results in Table 2 show that across both hard- and soft-intervention regimes, $\mathcal{F}$-FCI achieved precision gains exceeding 15%, and this advantage grew as the extent of post-treatment selection increased, confirming that explicitly modeling selection yields tangible benefits for structure recovery.

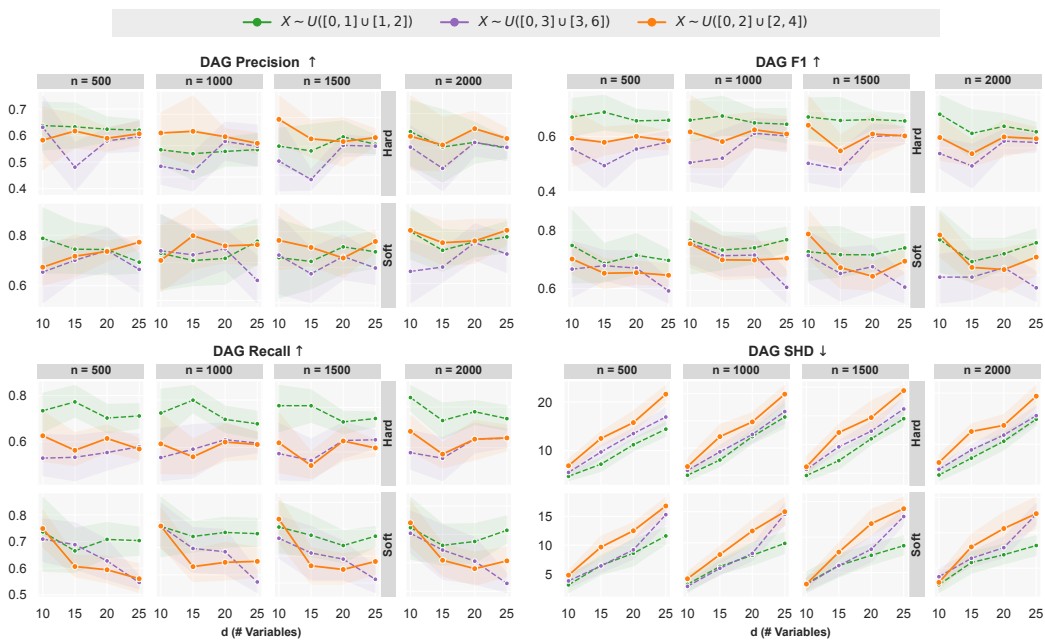

Figure 12: Robustness analysis of $\mathcal{F}$-FCI on $X$ with different noise levels under four metrics: DAG Precision, DAG $F_1$, DAG Recall, and DAG SHD (Structural Hamming Distance). All values are averaged over 10 runs with different random seeds. Error bars represent the 95% confidence interval.

Table 2: Comparison of $\mathcal{F}$-FCI and FCI-INTERVEN on graphs with 20 variables, 2–3 latent confounders, and 5–6 randomly chosen selection variables. All values are averaged over 10 runs.

| | $\mathcal{F}$-FCI | | | | FCI-INTERVEN | | | |
|---|---|---|---|---|---|---|---|---|
| $n$ | Precision | SHD | F1 | Recall | Precision | SHD | F1 | Recall |
| | | | | Hard intervention | | | | |
| 500 | $60.1 \pm 0.6$ | $12.6 \pm 1.8$ | $56.7 \pm 0.2$ | $55.7 \pm 1.0$ | $46.7 \pm 0.1$ | $16.3 \pm 3.2$ | $54.8 \pm 0.1$ | $69.0 \pm 2.1$ |
| 1500 | $62.1 \pm 0.6$ | $12.4 \pm 4.2$ | $58.2 \pm 0.7$ | $57.1 \pm 2.1$ | $46.8 \pm 0.1$ | $16.5 \pm 3.6$ | $56.8 \pm 0.2$ | $74.5 \pm 2.2$ |
| 2000 | $64.2 \pm 0.4$ | $10.9 \pm 4.7$ | $64.7 \pm 0.5$ | $66.4 \pm 1.3$ | $43.7 \pm 0.1$ | $17.2 \pm 6.7$ | $55.2 \pm 0.1$ | $77.8 \pm 2.1$ |
| | | | | Soft intervention | | | | |
| 500 | $67.6 \pm 0.6$ | $9.8 \pm 3.9$ | $63.0 \pm 0.8$ | $61.6 \pm 1.0$ | $49.0 \pm 0.2$ | $14.2 \pm 1.6$ | $54.1 \pm 0.3$ | $62.2 \pm 1.3$ |
| 1500 | $68.7 \pm 0.4$ | $8.7 \pm 3.2$ | $69.1 \pm 0.4$ | $70.2 \pm 0.4$ | $49.8 \pm 0.6$ | $14.1 \pm 5.49$ | $58.3 \pm 0.3$ | $72.2 \pm 1.1$ |
| 2000 | $70.3 \pm 1.5$ | $8.6 \pm 7.2$ | $68.8 \pm 0.9$ | $68.9 \pm 0.7$ | $48.1 \pm 0.4$ | $14.4 \pm 5.0$ | $57.1 \pm 0.2$ | $72.1 \pm 0.6$ |

To further assess the robustness of $\mathcal{F}$-FCI in more challenging and realistic settings, we conducted two stress tests: (i) richer nonlinear mechanisms and (ii) alternative noise families. For the nonlinearity test, in each run the structural function $f$ on every edge was sampled uniformly from

$$\mathcal{F} = \left\{ \sin(\pi x) + 0.2\sin(2\pi x), \ x^2, \ \tanh(x), \ x, \ x\, e^{-x^2/2}, \ \log\left(1 + e^{2x-1}\right) + 0.05\, x^2 \right\}.$$

Experimental results in Table 3 show a modest decline in *precision* as functional complexity increases, while overall performance remains competitive. For the noise test, we replaced *Uniform* noise with *Laplace* and *Gumbel* noises; for each variable in $X$, the location and scale were drawn independently as $\mu \sim \mathcal{U}(0,3)$ and $s \sim \mathcal{U}(2,4)$ (Laplace: $s = b$; Gumbel: $s = \beta$). Table 4 reports the experimental results of $\mathcal{F}$-FCI with *Laplace* noise; Table 5 reports the results with *Gumbel* noise. Across these noise families, $\mathcal{F}$-FCI's performance remains *stable* and in some settings *slightly improves*, indicating robustness to heavy-tailed and asymmetric error distributions.

## D.3 REAL-WORLD

The Norman dataset comprises gene expression profiles of 91,205 lung epithelial cells measured across 5,045 genes. Among these, 7,353 samples correspond to purely observational data (control group), while the remaining samples are derived from gene perturbation experiments. In total, 105

Table 3: Experimental results of $\mathcal{F}$-FCI under more complex nonlinearities on the graph with 15 variables. All values are averaged over 10 runs.

| | $\mathcal{F}$-FCI | | | | $\mathcal{F}$-FCI (complex non-linearities) | | | |
|---|---|---|---|---|---|---|---|---|
| $n$ | Precision | SHD | F1 | Recall | Precision | SHD | F1 | Recall |
| | Hard intervention | | | | | | | |
| 500 | $61.7 \pm 1.0$ | $12.5 \pm 5.9$ | $57.9 \pm 0.7$ | $56.3 \pm 1.4$ | $57.9 \pm 2.2$ | $8.6 \pm 4.6$ | $55.8 \pm 0.4$ | $55.5 \pm 0.3$ |
| 1500 | $62.6 \pm 1.1$ | $12.1 \pm 5.5$ | $57.4 \pm 0.7$ | $53.8 \pm 0.9$ | $56.3 \pm 0.9$ | $8.4 \pm 4.4$ | $58.2 \pm 0.9$ | $60.5 \pm 1.0$ |
| 2000 | $60.9 \pm 1.6$ | $12.5 \pm 6.6$ | $57.8 \pm 0.8$ | $56.8 \pm 1.0$ | $57.6 \pm 1.1$ | $8.3 \pm 7.4$ | $61.6 \pm 1.2$ | $67.6 \pm 1.9$ |
| | Soft intervention | | | | | | | |
| 500 | $72.7 \pm 1.6$ | $9.5 \pm 8.3$ | $65.7 \pm 0.6$ | $60.5 \pm 0.3$ | $64.0 \pm 2.6$ | $7.4 \pm 8.2$ | $62.9 \pm 1.6$ | $63.7 \pm 2.1$ |
| 1500 | $75.6 \pm 0.7$ | $7.9 \pm 4.3$ | $70.3 \pm 0.8$ | $66.4 \pm 1.2$ | $66.8 \pm 2.6$ | $6.1 \pm 9.9$ | $69.5 \pm 1.6$ | $73.3 \pm 1.2$ |
| 2000 | $75.7 \pm 1.8$ | $8.0 \pm 6.8$ | $72.2 \pm 0.6$ | $70.9 \pm 0.8$ | $71.4 \pm 1.2$ | $5.3 \pm 7.0$ | $73.4 \pm 1.2$ | $75.9 \pm 1.6$ |

Table 4: Comparison of $\mathcal{F}$-FCI under Uniform noise vs. Laplace noise on graph with 15 variables. All values are averaged over 10 runs.

| | Uniform | | | | Laplace | | | |
|---|---|---|---|---|---|---|---|---|
| $n$ | Precision | SHD | F1 | Recall | DAG Precision | DAG SHD | DAG F1 | DAG Recall |
| | Hard intervention (15 variables) | | | | | | | |
| 500 | $61.7 \pm 1.0$ | $12.5 \pm 5.9$ | $57.9 \pm 0.7$ | $56.3 \pm 1.4$ | $59.9 \pm 1.5$ | $8.6 \pm 5.6$ | $58.6 \pm 1.1$ | $58.7 \pm 1.6$ |
| 1500 | $62.6 \pm 1.1$ | $12.1 \pm 5.5$ | $57.4 \pm 0.7$ | $53.8 \pm 0.9$ | $64.1 \pm 2.4$ | $7.7 \pm 6.6$ | $62.3 \pm 1.5$ | $63.0 \pm 2.2$ |
| 2000 | $60.9 \pm 1.6$ | $12.5 \pm 6.6$ | $57.8 \pm 0.8$ | $56.8 \pm 1.0$ | $61.7 \pm 1.9$ | $7.7 \pm 7.6$ | $60.6 \pm 1.7$ | $60.4 \pm 2.2$ |
| | Soft intervention (15 variables) | | | | | | | |
| 500 | $72.7 \pm 1.6$ | $9.5 \pm 8.3$ | $65.7 \pm 0.6$ | $60.5 \pm 0.3$ | $69.8 \pm 2.2$ | $9.1 \pm 4.1$ | $59.2 \pm 1.2$ | $57.3 \pm 1.6$ |
| 1500 | $75.6 \pm 0.7$ | $7.9 \pm 4.3$ | $70.3 \pm 0.8$ | $66.4 \pm 1.2$ | $72.7 \pm 1.6$ | $7.4 \pm 4.8$ | $65.3 \pm 0.6$ | $63.7 \pm 0.7$ |
| 2000 | $75.7 \pm 1.8$ | $8.0 \pm 6.8$ | $72.2 \pm 0.6$ | $70.9 \pm 0.8$ | $73.4 \pm 1.8$ | $6.7 \pm 4.0$ | $67.5 \pm 0.9$ | $63.5 \pm 0.9$ |

Table 5: Comparison of $\mathcal{F}$-FCI under Uniform noise vs. Gumbel noise on graph with 15 variables. All values are averaged over 10 runs.

| | Uniform | | | | Gumbel | | | |
|---|---|---|---|---|---|---|---|---|
| $n$ | Precision | SHD | F1 | Recall | DAG Precision | DAG SHD | DAG F1 | DAG Recall |
| | Hard intervention (15 variables) | | | | | | | |
| 500 | $61.7 \pm 1.0$ | $12.5 \pm 5.9$ | $57.9 \pm 0.7$ | $56.3 \pm 1.4$ | $60.9 \pm 0.8$ | $9.6 \pm 4.0$ | $55.5 \pm 0.9$ | $52.6 \pm 1.6$ |
| 1500 | $62.6 \pm 1.1$ | $12.1 \pm 5.5$ | $57.4 \pm 0.7$ | $53.8 \pm 0.9$ | $57.6 \pm 0.8$ | $9.5 \pm 5.3$ | $57.0 \pm 1.1$ | $57.3 \pm 1.8$ |
| 2000 | $60.9 \pm 1.6$ | $12.5 \pm 6.6$ | $57.8 \pm 0.8$ | $56.8 \pm 1.0$ | $61.9 \pm 1.8$ | $9.1 \pm 9.9$ | $62.7 \pm 1.5$ | $65.5 \pm 2.2$ |
| | Soft intervention (15 variables) | | | | | | | |
| 500 | $72.7 \pm 1.6$ | $9.5 \pm 8.3$ | $65.7 \pm 0.6$ | $60.5 \pm 0.3$ | $69.4 \pm 3.1$ | $7.4 \pm 3.4$ | $58.6 \pm 1.2$ | $52.6 \pm 1.4$ |
| 1500 | $75.6 \pm 0.7$ | $7.9 \pm 4.3$ | $70.3 \pm 0.8$ | $66.4 \pm 1.2$ | $73.8 \pm 1.7$ | $6.7 \pm 4.2$ | $62.8 \pm 1.8$ | $61.6 \pm 3.0$ |
| 2000 | $75.7 \pm 1.8$ | $8.0 \pm 6.8$ | $72.2 \pm 0.6$ | $70.9 \pm 0.8$ | $76.9 \pm 2.6$ | $6.0 \pm 4.0$ | $66.4 \pm 1.6$ | $62.0 \pm 3.1$ |

genes were perturbed individually. Following the procedure of Algorithm 1, a parallel algorithm Fast-FCI (Ramsey et al., 2017) with BIC score is first utilized to get the skeleton over 5045 genes, due to the large number of genes. Then, the rules of CI patterns are utilized to orient the skeleton among perturbed genes.

To evaluate the effectiveness of $\mathcal{F}$-FCI in identifying causal relations, latent confounders, and post-treatment selection, we conducted experiments on gene perturbation (interventional) data, with the results summarized in Figure 13. Several identified causal links are consistent with established biological evidence, including $IGDCC3 \rightarrow FEV$ is aligned with biological evidence provided by ARCHS4 (Lachmann et al., 2018) and Enrichr (Chen et al., 2013), $CEBPA \rightarrow FEV$ is supported by Enrichr, $SET \rightarrow FOXL2$ and $SET \rightarrow KMT2A$ are supported by GEO (Clough & Barrett, 2016), $SET \rightarrow PRDM1$ is supported by ChEA2022 (Keenan et al., 2019) and ENCODE-TF-CHIP-seq (Consortium et al., 2012), $HOXC13 \rightarrow SET$ is supported by ChEA2022, $ZC3HAV1 \rightarrow TSC22D1$ is supported by Enrichr, $LYL1 \rightarrow KLF1$ is supported by ChEA2022 and ARCHS4, $CELF2 \rightarrow FOXL2$ is supported by ARCHS4, GEO, and Enrichr. Moreover, $\mathcal{F}$-FCI flags several genes, such as ZNF318, CDKN1C, CDKN1A, and RREB1, as being affected by post-treatment selection. Here, post-treatment selection refers to the scRNA-seq quality-control (QC) filtering step

that retains only transcriptionally active, viable cells, typically those with high nFeature and high nCount, together with a low fraction of mitochondrial transcripts (percent.mt). To contextualize this selection effect, prior work links CDKN1A (p21) and CDKN1C (p57) to DNA-damage responses and cell-cycle inhibition. CDKN1A is a canonical DNA-damage–responsive CDK inhibitor that enforces cell-cycle arrest and is implicated in growth suppression, transcriptional reprogramming, and apoptosis under genotoxic stress (El-Deiry et al., 1993; Harper et al., 1993). CDKN1C is another CDK inhibitor (acting primarily at G1) that negatively regulates proliferation and helps maintain quiescent/non-proliferative states (Matsuoka et al., 1995; Creff & Besson, 2020). Cells exhibiting elevated CDKN1A/CDKN1C activity are therefore plausibly less transcriptionally active, consistent with lower nFeature/nCount and a higher likelihood of being excluded by QC. In addition, RREB1 and ZNF318 have been associated with mitochondrial-related programs. RREB1 is a Ras-responsive transcription factor that has been implicated in the regulation of nuclear-encoded mitochondrial respiratory-chain components (Han et al., 2024), while ZNF318 (also reported as ZFP318 in some contexts) is linked to transcription/splicing regulation and has been connected to mitochondrial gene-expression programs in specific immune states (e.g., memory B-cell–related programs) (Wang et al., 2024). Together, these associations are consistent with QC-based post-treatment selection that is sensitive to mitochondrial transcript burden and overall transcriptional activity.

## E  DISCUSSION

**Pre-treatment selection vs. biological constraints vs. post-treatment selection.**

CDIS Dai et al. (2025) addresses *pre-treatment selection*, where samples are filtered *before* interventions (e.g., screening drug-trial participants prior to treatment assignment). GISL Luo et al. (2025) identifies biological constraints, which happen before intervention and continuously filter out non-viable cells. However, $\mathcal{F}$-FCI targets *post-treatment selection*, where samples are retained *after* treatment/measurement (e.g., in gene expression analysis, cells that pass quality control in both observational and interventional groups are included in the dataset for analysis). This difference in when intervention interacts with selection yields opposite cross-setting invariant/variability patterns for variable pairs under selection: under pre-treatment selection, *marginal* distributions are invariant, while *conditional* distributions change; under biological constraints, both *marginal* and *conditional* distributions change; under post-treatment selection, *marginal* distributions change (via collider/selection effects) while the *conditional* distribution remains invariant. Methodologically, to model the different interaction patterns between intervention and selection, CDIS and GISL model interventions via an *interventional twin graph*, whereas $\mathcal{F}$-FCI uses an *augmented DAG* with an explicit selection node $S$ and intervention indicators to leverage these invariance/variability patterns. Theoretically, $\mathcal{F}$-FCI is sound and complete, and characterizes a finer-grained Markov equivalence class.

## F  THE USE OF LARGE LANGUAGE MODELS

An LLM was used to refine writing for clarity and readability but did not contribute to the research design, experiment, or analysis. All intellectual work was independently conducted by the authors, and any suggestions from the LLM were critically evaluated before use. The authors bear full responsibility for the research, and the LLM is not listed as a contributor or author.

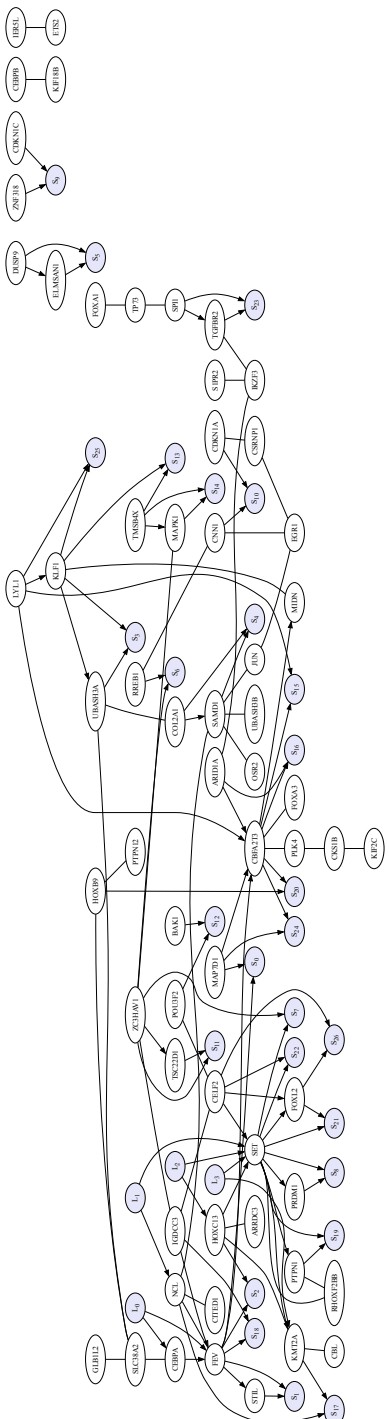

Figure 13: The causal structure among genes with interventional data identified by $\mathcal{F}$-FCI.

