# OpenReview forum: "Characterization and Learning of Causal Graphs with Latent Confounders and Post-treatment Selection from Interventional Data"
_ICLR.cc/2026/Conference — ICLR 2026 Poster_

### Official Review · Reviewer_SifF · 2025-10-24

**Soundness:** 4
**Presentation:** 3
**Contribution:** 4
**Rating:** 8
**Confidence:** 4

**Summary:**

This paper addresses an important problem: interventional causal discovery under post-treatment selection. The authors systematically analyze how post-treatment selection affects causal identifiability and summarize invariant and variant distributional patterns that arise in such settings. Building upon existing interventional frameworks, they propose a novel causal formulation that explicitly models post-treatment selection, i.e., FI-Markov, F-PAG and  a new algorithm F-FCI. Experiments on both synthetic and real-world datasets demonstrate the effectiveness and robustness of the proposed approach.

**Strengths:**

1.	The characterization of invariant/variant rules under post-treatment selection and the introduction of the causal formulation are novel.
2.	The paper provides a solid theoretical foundation for F-FCI.

**Weaknesses:**

1.	I did not observe any discussion in the paper regarding how F-FCI handles Type II inducing nodes.
2.	Figure 10 shows that GIES outperforms F-FCI under hard interventions in terms of F1 and recall, but the paper does not analyze why this occurs. A brief discussion would help clarify the underlying cause of this performance gap.
3.	There is a typo in Figures 10 and 11, R-FCI should be corrected to F-FCI.
4.	It would strengthen the paper if the authors could include a case study showing which edges existing causal discovery algorithms falsely infer due to post-treatment effects, and how F-FCI correctly identifies and removes these spurious edges.

**Questions:**

1.	How does F-FCI handle Type II inducing nodes?
2.	Why does GIES outperform F-FCI under hard interventions in terms of F1 and recall?
3.	Should “R-FCI” in Figures 10 and 11 be corrected to “F-FCI”?
4.	Can the authors provide a case study showing spurious edges caused by post-treatment selection that existing algorithms falsely discover, and that F-FCI correctly identifies?

---

> ### Author Response · Authors · 2025-11-20
>
> We sincerely appreciate the reviewer's constructive comments and helpful feedback. Please see below for our response.
>
> **Q1:**  How does $\mathcal{F}$-FCI handle Type II inducing nodes?
>
> **A1:** As discussed in our limitations (Line 1114), $\mathcal{F}$-FCI **cannot** handle inducing paths that are **composed solely of Type II inducing nodes**. In particular, obtaining a fine-grained interventional Markov equivalence that distinguishes complex structures such as the motivating example in Figure 1 **requires the presence of at least one Type I inducing node** (see Line 410–414). When the inducing path contains only Type II nodes, the interventions on inducing nodes can not help identify the concrete causal structure.
>
> For example, considering $X_1 \rightarrow X_2 \rightarrow X_3$, $X_1 \rightarrow S \leftarrow X_2$, $X_2 \leftarrow L \rightarrow X_3$, where $X_2$ is the type II inducing node. Even under interventions on $X_2$, changes at $X_2$ can propagate to $X_3$ via $X_2 \rightarrow X_3$. Consequently, the observed signals are compatible with **both** the presence and the absence of a direct edge $X_1 \to X_3$; $\mathcal{F}$-FCI cannot decide between these alternatives in the absence of a Type I inducing node.
>
> ---
>
> **Q2:**   Why does GIES outperform F-FCI under hard interventions in terms of F1 and recall?
>
> **A2:**  Because GIES does not model latent confounders or post-treatment selection, it tends to include edges wherever conditional dependencies are detected, producing comparatively dense graphs. This leads to **high recall but lower precision** (shown in Figures 6 and 10). Moreover, GIES is tailored to **hard interventions**. In **small-graph, low-sample (≈500)** cases, higher inference uncertainty combined with a limited edge space means false positives depress precision less severely; consequently, GIES’s high-recall tendency may yield a good F1 score.
>
> ---
>
> **Q3:**  Should “R-FCI” in Figures 10 and 11 be corrected to “F-FCI”?
>
> **A3:** Yes. Thanks for pointing it out. We have corrected it in the revision.
>
>  ---
>
> **Q4:**  Can the authors provide a case study showing spurious edges caused by post-treatment selection that existing algorithms falsely discover, and that F-FCI correctly identifies?
>
> **A4:** Considering motivation example Figure 1(a) as a case study. The true graph $X_1 \rightarrow X_3 \leftarrow L \rightarrow X_2$,  with post-treatment selection $X_1 \rightarrow S \leftarrow X_3$. We only observe the data over $X=\{X_1, X_2, X_3\}$ with $S$=1 (after selection). Conditioning on $S$ opens the collider at $S$, inducing an extra **spurious dependence between $X_1$ and $X_3$**; because $S$ is a descendant of the collider $X_1 \rightarrow X_3 \leftarrow L$, it also opens $X_1 \rightarrow X_3  \leftarrow L  \rightarrow X_2$, creating a **spurious dependence between** $X_1$ **and** $X_2$.
>
>
> - Baselines:
> FCI-INTERVEN and CDIS, which can only handle latent confounders and can not handle post-treatment selection, then result in $X_1 \mathrel{-}\circ X_3 \circ\rightarrow X_2$, $X_1 \rightarrow X_2$. For other baselines, as they can not handle both latent confounders and post-treatment selection, result in $X_1 \circ\mathrel{-}\circ X_3\circ\mathrel{-}\circ X_2$, $X_1 \rightarrow X_2$.
>
> - $\mathcal{F}$-FCI:
> By modeling post-treatment selection and using changes induced by intervention, $\mathcal{F}$-FCI correctly recovers $X_1 \rightarrow X_3$, recognizes latent confounders between $X_3$ and $X_2$, **removes the spurious edge** $X_1 \mathrel{-} X_2$, and identifies $X_1 \rightarrow S \leftarrow X_3$.

---

> > ### Comment · Reviewer_SifF · 2025-11-26
> >
> > Thank you for your detailed and thoughtful replies to my comments. The clarifications effectively address the concerns I raised, and I appreciate the effort you put into strengthening the manuscript. I find the overall quality of the work solid, the motivation well grounded, and the technical development rigorous. I will maintain my original positive rating.

---

> > > ### Author Response · Authors · 2025-11-26
> > >
> > > We sincerely appreciate your thoughtful feedback and recognition of our work. Thank you for your time.

---

### Official Review · Reviewer_1GEE · 2025-10-24

**Soundness:** 3
**Presentation:** 3
**Contribution:** 4
**Rating:** 8
**Confidence:** 2

**Summary:**

The authors investigate the issue of post-treatment selection in the context of causal discovery. They develop a thorough framework for characterizing post-treatment selection and introduce a new class of observationally equivalent causal structures in this context.  Furthermore, a new algorithm based on their framework is discovered and  investigated theoretically (soundness and completeness) and experimentally on synthetic and real-world data.

**Strengths:**

The problem that the paper addresses is well motivated and I agree that it is an overlooked yet critical issue. All claims are well supported by evidence.

I am also not aware that this problem has been addressed (in this way) before.

The conceptual and theoretical analysis of post-treatment selection is rigorous and extensive. The paper is clearly written, well motivated and supported by effective figures.

The proposed algorithm is reasonable and the provided experimental results seem very sound and support the contribution by the paper.

**Weaknesses:**

The experiments are somewhat limited:
(a) the synthetic data is rather simple, it would be interesting to see how the method performs with different non-linearities, forms of noise and larger graphs; including and ablation over those factors. It would be especially interesting to see how well the proposed method can perform on realistic simulators introduced by Robertson et al. (https://arxiv.org/abs/2506.06039).


The discussion of limitations and future work for the method is fairly short and could benefit from being extended.


NIT: There is an extra space at the end of line 166

**Questions:**

Please feel free to reply to the weaknesses above.

---

> ### Author Response · Authors · 2025-11-20
> **Response to Reviewer 1GEE (Part 1/2)**
>
> We sincerely appreciate your insightful suggestions and valuable feedback, which have enabled us to strengthen our method through a more comprehensive evaluation. Please see below for our response.
>
> **Q1:** The synthetic data is rather simple; it would be interesting to see how the method performs with different non-linearities, forms of noise, and larger graphs, including an ablation over those factors.
>
> **A1:** In light of your suggestion, we have conducted comprehensive experiments to evaluate the effectiveness of $\mathcal{F}$-FCI in more complex and realistic settings as follows:
>
> **Different non-linearities.**
>
> We conducted experiments to evaluate the effectiveness of  $\mathcal{F}$-FCI  under more complex non-linearities, the structural function function $f$ is randomly drawn from $F =\{ \sin(\pi x) + 0.2\,\sin(2\pi x), x^{2}, \tanh(x), x, x\,e^{-x^{2}/2}, \log\!\big(1 + e^{2x-1}\big) + 0.05\,x^{2}\}$. **Experimental results show that $\mathcal{F}$-FCI's performance experiences a slight decline in Precision as the complexity increases in non-linearities.**
>
>
> Experimental setting: hard intervention, 15 variables.
>
> | | **$\mathcal{F}$-FCI** | | | | **$\mathcal{F}$-FCI** (complex non-linearities) | | | |
> |---|---|---|---|---|---|---|---|---|
> |  n | Precision | SHD | F1 | Recall | Precision | SHD | F1 |  Recall |
> | 500  | 61.7 ± 1.0 | 12.5 ± 5.9 | 57.9 ± 0.7 | 56.3 ± 1.4 | 57.9 ± 2.2 | 8.6 ± 4.6 | 55.8 ± 0.4 | 55.5 ± 0.3 |
> | 1500 | 62.6 ± 1.1 | 12.1 ± 5.5 | 57.4 ± 0.7 | 53.8 ± 0.9 | 56.3 ± 0.9 | 8.4 ± 4.4 | 58.2 ± 0.9 | 60.5 ± 1.0 |
> | 2000 | 60.9 ± 1.6 | 12.5 ± 6.6 | 57.8 ± 0.8 | 56.8 ± 1.0 | 57.6 ± 1.1 | 8.3 ± 7.4 | 61.6 ± 1.2 | 67.6 ± 1.9 |
>
>
> Experimental setting: soft intervention, 15 variables.
>
> | | **$\mathcal{F}$-FCI** | | | | **$\mathcal{F}$-FCI** (complex non-linearities) | | | |
> |---|---|---|---|---|---|---|---|---|
> |  n | Precision | SHD | F1 | Recall | DAG Precision | DAG SHD | DAG F1 | DAG Recall |
> | 500  | 72.7 ± 1.6 | 9.5 ± 8.3 | 65.7 ± 0.6 | 60.5 ± 0.3 | 64.0 ± 2.6 | 7.4 ± 8.2 | 62.9 ± 1.6 | 63.7 ± 2.1 |
> | 1500 | 75.6 ± 0.7 | 7.9 ± 4.3 | 70.3 ± 0.8 | 66.4 ± 1.2 | 66.8 ± 2.6 | 6.1 ± 9.9 | 69.5 ± 1.6 | 73.3 ± 1.2 |
> | 2000 | 75.7 ± 1.8 | 8.0 ± 6.8 | 72.2 ± 0.6 | 70.9 ± 0.8 | 71.4 ± 1.2 | 5.3 ± 7.0 | 73.4 ± 1.2 | 75.9 ± 1.6 |
>
> **Different forms of noise**
>
> We conducted experiments to evaluate the effectiveness of  $\mathcal{F}$-FCI  under **Laplace** and **Gumbel** noise. In each run, noise parameters are drawn independently with location $\mu\sim U(0,3)$ and scale $s \sim \mathcal{U}(2,4)$ (for Laplace $s=b$; for Gumbel $s=\beta$). **Experimental results indicate that $\mathcal{F}$-FCI's performance remains stable and even slightly improves in some settings across these noise families.**
>
> Experimental setting: hard intervention, 15 variables.
>
> | | **Uniform** | | | | **Laplace** | | | |
> |---|---|---|---|---|---|---|---|---|
> |  n | Precision | SHD | F1 | Recall | Precision | SHD |  F1 | Recall |
> | 500  | 61.7 ± 1.0 | 12.5 ± 5.9 | 57.9 ± 0.7 | 56.3 ± 1.4 | 59.9 ± 1.5 | 8.6 ± 5.6 | 58.6 ± 1.1 | 58.7 ± 1.6 |
> | 1500 | 62.6 ± 1.1 | 12.1 ± 5.5 | 57.4 ± 0.7 | 53.8 ± 0.9 | 64.1 ± 2.4 | 7.7 ± 6.6 | 62.3 ± 1.5 | 63.0 ± 2.2 |
> | 2000 | 60.9 ± 1.6 | 12.5 ± 6.6 | 57.8 ± 0.8 | 56.8 ± 1.0 | 61.7 ± 1.9 | 7.7 ± 7.6 | 60.6 ± 1.7 | 60.4 ± 2.2 |
>
> Experimental setting: soft intervention, 15 variables.
>
> | | **Uniform** | | | | **Laplace** | | | |
> |---|---|---|---|---|---|---|---|---|
> |  n | Precision | SHD | F1 | Recall | Precision | SHD |  F1 |  Recall |
> | 500  | 72.7 ± 1.6 | 9.5 ± 8.3 | 65.7 ± 0.6 | 60.5 ± 0.3 | 69.8 ± 2.2 | 9.1 ± 4.1 | 59.2 ± 1.2 | 57.3 ± 1.6 |
> | 1500 | 75.6 ± 0.7 | 7.9 ± 4.3 | 70.3 ± 0.8 | 66.4 ± 1.2 | 72.7 ± 1.6 | 7.4 ± 4.8 | 65.3 ± 0.6 | 63.7 ± 0.7 |
> | 2000 | 75.7 ± 1.8 | 8.0 ± 6.8 | 72.2 ± 0.6 | 70.9 ± 0.8 | 73.4 ± 1.8 | 6.7 ± 4.0 | 67.5 ± 0.9 | 63.5 ± 0.9 |
>
> Experimental setting: hard intervention, 15 variables.
> | | **Uniform** | | | | **Gumbel** | | | |
> |---|---|---|---|---|---|---|---|---|
> |  n | Precision | SHD | F1 | Recall | Precision |  SHD | F1 |  Recall |
> | 500  | 61.7 ± 1.0 | 12.5 ± 5.9 | 57.9 ± 0.7 | 56.3 ± 1.4 | 60.9 ± 0.8 | 9.6 ± 4.0 | 55.5 ± 0.9 | 52.6 ± 1.6 |
> | 1500 | 62.6 ± 1.1 | 12.1 ± 5.5 | 57.4 ± 0.7 | 53.8 ± 0.9 | 57.6 ± 0.8 | 9.5 ± 5.3 | 57.0 ± 1.1 | 57.3 ± 1.8 |
> | 2000 | 60.9 ± 1.6 | 12.5 ± 6.6 | 57.8 ± 0.8 | 56.8 ± 1.0 | 61.9 ± 1.8 | 9.1 ± 9.9 | 62.7 ± 1.5 | 65.5 ± 2.2 |
>
> Experimental setting: soft intervention, 15 variables.
>
> | | **Uniform** | | | | **Gumbel** | | | |
> |---|---|---|---|---|---|---|---|---|
> |  n | Precision | SHD | F1 | Recall | Precision | SHD |  F1 | Recall |
> | 500  | 72.7 ± 1.6 | 9.5 ± 8.3 | 65.7 ± 0.6 | 60.5 ± 0.3 | 69.4 ± 3.1 | 7.4 ± 3.4 | 58.6 ± 1.2 | 52.6 ± 1.4 |
> | 1500 | 75.6 ± 0.7 | 7.9 ± 4.3 | 70.3 ± 0.8 | 66.4 ± 1.2 | 73.8 ± 1.7 | 6.7 ± 4.2 | 62.8 ± 1.8 | 61.6 ± 3.0 |
> | 2000 | 75.7 ± 1.8 | 8.0 ± 6.8 | 72.2 ± 0.6 | 70.9 ± 0.8 | 76.9 ± 2.6 | 6.0 ± 4.0 | 66.4 ± 1.6 | 62.0 ± 3.1 |

---

> > ### Author Response · Authors · 2025-11-20
> > **Response to Reviewer 1GEE (Part 2/2)**
> >
> > **Larger graphs.**
> >
> > We conducted experiments to evaluate the effectiveness of  $\mathcal{F}$-FCI on larger graphs (Line 999 Figure 11). **Experimental results indicate that $\mathcal{F}$-FCI's performance remains stable and even improves on larger graphs, owing to the greater sparsity of the corresponding graphs.**
> >
> > Experimental setting: soft intervention, 1500 samples.
> >
> > | Number of variables | Precision | SHD | F1 | Recall |
> > |-----------------|-----------|-----|----|--------|
> > | 15 | 75.6 $\pm$ 0.7 | 7.9 $\pm$ 4.3 | 70.3 $\pm$ 0.8 | 66.4 $\pm$ 1.2 |
> > | 25 | 77.3 $\pm$ 0.3 | 14.0 $\pm$ 6.0 | 72.1 $\pm$ 0.3 | 68.0 $\pm$ 0.6 |
> > | 50 | 78.9 $\pm$ 0.2 | 12.6 $\pm$ 7.4 | 71.4 $\pm$ 0.7 | 67.3 $\pm$ 1.1 |
> >
> > ---
> >
> > **Q2:** It would be especially interesting to see how well the proposed method can perform on realistic simulators introduced by Robertson et al.
> >
> > **A2:** **Following your suggestion (Robertson et al.’s setting),** we ran the experiments listed above under **more complex structural functions** and **multiple noise families** to better approximate realistic scenarios. However, **RealCause (used in Robertson et al.) [2]** is built for **causal effect estimation**, not **causal discovery**, and is therefore not suitable for evaluating $\mathcal{F}$-FCI. The key gaps are:
> >
> > 1.  **Objective & ground truth differ.** RealCause provides ground truth **effects** (ATE/CATE), not a ground-truth **graph** (edges/orientations). $\mathcal{F}$-FCI must be evaluated on structural criteria (adjacency/arrowheads/SHD), which RealCause does not supply.
> >
> > 2.  **Data modality: no interventional environments over X**. **RealCause generates a single observational distribution P(W,T,Y)** and, at best, counterfactual outcomes under $\mathrm{do}$(T=t). $\mathcal{F}$-FCI relies on **multiple interventional environments** with **known targets** $I$ and uses **invariance/variability across environments** to orient edges. RealCause does not provide interventional datasets on arbitrary variables $X_i$ (only on the treatment T), nor the interventional data needed by $\mathcal{F}$-FCI.
> >
> > 3.  **Assumptions incompatible with our setting.** RealCause is built to **control confounding** for effect estimation: it assumes observed covariates W suffice (ignorability), **no post-treatment selection**, and focuses on  $T \rightarrow Y$(effect). In contrast, $\mathcal{F}$-FCI explicitly **models latent confounders** and **post-treatment selection** (selection node S), and its identifiability/edge orientation crucially uses these structures. RealCause lacks both.
> >
> > 4.  **Variable scope and mechanisms.** RealCause treats W as covariates with **no specified causal relations among the** W. $\mathcal{F}$-FCI requires a **multivariate causal graph** over variables $X$ with edges among them, plus known intervention targets. RealCause only specifies $P(T \mid W)$ and $P(Y \mid T,W)$; it does **not** define mechanism changes for general $X_i$.
> >
> > [1] Do-PFN: In-Context Learning for Causal Effect Estimation, 2025.
> >
> > [2] Realcause: Realistic causal inference benchmarking. 2020
> >
> > ---
> >
> > **Q3:** The discussion of limitations and future work for the method is fairly short and could benefit from being extended.
> > **A3:** **We have extended the discussion of limitations and future work in Line 1115** as follows:
> > Leveraging interventional data and explicitly modeling post-treatment selection and latent confounders done by $\mathcal{F}$-FCI, yields fine-grained Markov equivalence classes, thereby moving closer to the underlying true causal structures. However, as noted in **Step 2.2** (Line 374), when the intervention indicator $\psi$ points to a confounded variable, for example, $\psi_1 \rightarrow X_1 \leftarrow L \rightarrow X_2$ and $X_1$ is subject to selection, $X_1$ cannot be blocked by conditioning; consequently, the causal relation between $X_1$and $X_2$ remains unidentifiable (denoted $X_1 \circ\to X_2$). **Handling this special case without extra information is a direction for future work**. Moreover, the identification of direct causal links and selection structures depends critically on the presence of Type I  inducing nodes: $\mathcal{F}$-FCI cannot identify the causal structure along inducing paths composed solely of Type II inducing nodes. **Extending the framework to accommodate the only Type II inducing node setting is an important avenue for future research.**
> >
> > ---
> >
> > **Q4:** There is an extra space at the end of line 166.
> >
> > **A4:** Thanks for pointing it out. We have corrected it in the revision.

---

> > > ### Comment · Reviewer_1GEE · 2025-11-21
> > >
> > > Thank you for running those new experiments and for replying to my questions and concerns. I believe the new experiments help to further confirm experimentally that the proposed method works in more complex setups; this addresses my concern.
> > >
> > > As a sidenote, I believe that I made my earlier point about the Robertson et al. paper not sufficiently clear. I certainly would not expect you to run their evaluation pipeline and fully agree with all your reasons to not use RealCause.
> > > However, if I understand correctly, Robertson et al. propose a powerful simulator of SCMs, inspired by TabPFN, and then train a foundation model on the simulator of SCMs to learn how to estimate causal effects, which ultimately shows quite strong performance on real-world datasets.
> > > This made me think that data from their simulator of SCMs could be a nice source for abundant synthetic (yet somewhat realistic and challenging) testing data.
> > >
> > >
> > > I maintain my initial rating of 8 (accept).

---

> > > > ### Author Response · Authors · 2025-11-24
> > > >
> > > > We are so happy that your concerns have been addressed, and we sincerely appreciate your thoughtful and constructive feedback. Your comments have helped strengthen our work by encouraging a more comprehensive evaluation and providing valuable recognition of our contributions.
> > > >
> > > > Thank you for clarifying your point about Robertson et al. We agree that their simulator of SCMs is a powerful source of abundant, realistic, and challenging synthetic tasks, and their foundation-model approach is tailored to causal effect estimation. We followed their setup to generate synthetic data for evaluating $\mathcal{F}$-FCI, as outlined in our response to Q1.

---

### Official Review · Reviewer_eoQR · 2025-10-31

**Soundness:** 3
**Presentation:** 3
**Contribution:** 3
**Rating:** 8
**Confidence:** 4

**Summary:**

The authors present a new formulation for interventional causal discovery which allows for explicit modeling of so-called post-treatment selection and for considering the extent to which differential responses to interventions may distinguish causal relationships from selection patterns. They propose a new definition of Markov equivalence, called FI-Markov equivalence, to characterize equivalence classes for the new setting, and provide a new graphical representation for an FI-Markov equivalent class, called F-PAG. They then modify the canonical FCI learning algorithm to develop the F-FCI algorithm for learning F-PAGs and prove its correctness and completeness.

**Strengths:**

“Post-treatment selection bias” is a well-known phenomenon in empirical research and is an important issue in causal inference, with implications for research in medicine, the social sciences, and other fields. To my knowledge, this work presents the first systematic study of the interventional causal discovery, taking into account post-treatment selection. The introduced new concepts of Markov equivalence (FI-Markov equivalence) and graphical models for equivalence classes (F-PAGs) constitute an interesting contribution to the field. An important achievement is the development of a provably sound and complete algorithm, F-FCI, that enables the identification of cause-effect relationships, latent confounders, and post-treatment selection.

**Weaknesses:**

The relationship of this article to the paper introducing pre-treatemt selection framework presented at the ICLR by Dai et al. (2025) is not clear and should be further discussed.

**Questions:**

L. 106: Conditioning on S = 1 encodes selection for both pre-treatment selection
and post-treatment selection. Q: Why don't you differentiate between these types of selection? Later, e.g., L.187 you write: S = 1 indicates the presence of post-treatment selection.

Eq.(1): what does the subscript s in p_s^{(k)} mean? Moreover, in the first product the probability p^{(i)} does not make sense.

L. 134: Do you make any assumptions about the collection of intervention targets, other than that I^{(0)} is empty?

Def. 1 is not clear: explain I, {\mathcal I}, I^{(k)}.

L.213: improve the punctation.

L. 234: what do you mean by Aug_{I_K} ? Should be Aug_{I^{k}}?

L. 235: the corresponding augmented DAG --> you mean DAGs?

Definition 2 is not clear: what does X_i ∈ X_{\cup I} mean?

L. 332:   R-PAG --> F-PAG

---

> ### Author Response · Authors · 2025-11-20
> **Response to Reviewer eoQR (Part 1/2)**
>
> Thank you for the time dedicated to reviewing our paper. Your insightful comments and valuable feedback have greatly improved its clarity. Following your suggestions, we have carefully revised the manuscript. Please see our point-by-point responses below.
>
> **Q1:** The relationship of this article to the paper introducing the pre-treatment selection framework presented at the ICLR by Dai et al. (2025) is not clear and should be further discussed.
>
> **A1:** In the revised version, we discuss the differences between pre-treatment and post-treatment selection as follows:
> | Mehtod|CDIS (Dai et al.) | $\mathcal{F}$-FCI (Ours) |
> |---|----------|----------|
> |Selection type| **Pre-treatment selection** (selection occurs before intervention)|**Post-treatment selection** (selection occurs after treatment)|
> |Real-world example|Drug trial participants are screened/selected (patients of relevant disease) before treatment assignment| Both observational and interventional (treatment) samples are retained only if they meet quality control criteria|
> |Interaction with interventions|**Marginal distribtuion**: invariant after intervention; **Conditional distribution**: varies after intervention|**Marginal distribtuion**: varies after intervention; **Conditional distribution**: invariant after intervention|
> |Modeling intervention|Interventional twin graph| Augmented DAG|
>
> ---
>
> **Q2:** L. 106: Conditioning on $S$ = 1 encodes selection for both pre-treatment selection and post-treatment selection. Why don't you differentiate between these types of selection? Later, e.g., L.187, you write: $S$  = 1 indicates the presence of post-treatment selection.
>
> **A2:** **If one wishes to model both pre- and post-treatment selection simultaneously in the same structural causal model, we agree it is clearer to distinguish them explicitly. In our paper, we introduce $S$ in two scopes, and we have clarified this in the revision**.  At Line106, we follow the standard convention: $S$  is a generic selection variable in observational data (page 143 [1]), which can represent any selection mechanism, pre- or post-treatment (page 2 [2]). In the part where we analyze how interventions interact with selection (Line 187), we focus **specifically on post-treatment selection**, therefore, we instantiate $S$ =1 to denote _post-treatment_ selection.
>
> [1] Causal Discovery from Data in the Presence of Selection Bias. Proceedings of Machine Learning Research. 1995.
>
> [2] Endogenous selection bias: The problem of conditioning on a collider variable. Annual review of sociology. 2014
>
>
> ----
>
> **Q3:** Eq.(1): what does the subscript s in $p_{s}^{(k)}$ mean? Moreover, in the first product, the probability $p^{(i)}$ does not make sense.
>
> **A3:** The subsrcript s in $p_{s}^{(k)}$ indicate the joint interventional distribution after post-treatement selection, which is ultilized to distinguish the joint interventional distribution over $X$ without post-treatment selection ($p^{(k)}$) from the corresponding distribution with post-treatment selection ($p_s^{(k)}$).
>
> Thanks for pointing it out,  we have corrected $p^{(i)}$  to $p^{(k)}$ in the revision.
>
> ---
>
>
> **Q4:** L. 134: Do you make any assumptions about the collection of intervention targets, other than that $I^{(0)}$ is empty?
>
> **A4:**  Aside from including the observational baseline $I^{(0)}=\varnothing$, we assume the **target set for each experiment** $I$ is known (L.133), and that **the changes between two settings (observation-intervention or intervention-intervention) can be isolated**, i.e., when contrasting two settings used for orientation, they differ in exactly one intervention target. For example, if the k-th intervention target is $I^{(k)}= \\{\varnothing, \\{1,2\\}\\} $, when orienting the edge between $X_1$ and any other intervened variable $X_n$ ($n \notin I^{(k)}$), we cannot attribute the observed invariance/variability between different settings ($\varnothing$ vs. $\\{1,2\\}$) to $X_1\mathrel{-}X_n$ or $X_2\mathrel{-}X_n$, as both $X_1$ and $X_2$ vary across two settings. In contrast, comparing $\varnothing$ vs. $\\{1\\}$ or $\\{2\\}$ vs. $\\{1,2\\}$ isolates the change to $X_1$, enabling identifiable orientation between $X_1$ and $X_n$.

---

> > ### Author Response · Authors · 2025-11-20
> > **Response to Reviewer eoQR (Part 2/2)**
> >
> > **Q5:** Def. 1 is not clear: explain $I$, ${\mathcal I}$, $I^{(k)}$.
> >
> > **A5:** We have clarified them in the revision. $\mathcal{I}$ denotes the collection of intervention target sets used across experiments (as defined in Line 134). $I$ denotes a generic intervention target, and $I^{(k)}$ indicates the target set for the $k$-th intervention.
> >
> > ---
> >
> > **Q6:** L.213: improve the punctuation.
> >
> > **A6:** Thank you for pointing this out. In the revision, we corrected the punctuation at Line 213 by replacing the periods after items 1)–3) with colons, and ensured consistency throughout.
> >
> > ---
> >
> > **Q7:** L. 234: What do you mean by Aug_{I_K}? Should be Aug_{I^{k}}?
> >
> > **A7:** Yes, it should be $Aug_{I^{(k)}}$, indicating the augmented DAG for the $k$-th intervention. We have corrected it in the revision.
> >
> > ---
> >
> > **Q8:**  L. 235: the corresponding augmented DAG --> you mean DAGs?
> >
> > **A8:** Yes, we have corrected it in the revision to improve clarity.
> >
> > ---
> >
> > **Q9:** Definition 2 is not clear: what does $X_i \in X_{\cup I}$ mean?
> >
> > **A9:** $X_i \in X_{\cup \mathcal{I}}$ indicates the intervened variable, where $X_{\cup \mathcal{I}}$ indicates the set of all intervened variables.  For example, $\mathcal{I}$ = {$I^{(0)}$={0,1}, $I^{(1)}$={1,2}}, $X_{\cup \mathcal{I}}$= {$X_0$, $X_1$, $X_2$}. We have clarified this in the revision.
> >
> > ---
> >
> > **Q10:** L. 332: R-PAG --> F-PAG
> >
> > **A10:** Yes, we have corrected it in the revision.

---

### Official Review · Reviewer_uvY9 · 2025-11-09

**Soundness:** 4
**Presentation:** 3
**Contribution:** 3
**Rating:** 6
**Confidence:** 4

**Summary:**

The paper addresses learning a Markov Equivalence class amongst causal graphs what admit latent variables in the Semi-Markovian model, interventional distributions based on soft mechanism changes and unknown post treatment selection variables. It is assumed that the identity of the interventions are known.

Paper makes a case that equivalence classes under interventions and under selection for semi markovian models has been characterized. But the Markov equivalence classes become very different when both are present - interventions and selection variables on post interventional distributions. Authors define a generalized notion of Markov equivalence based on marginal invariance tests, conditional invariance tests and also conditional independence under selection.

Authors characterize the Markov Equivalence class under this new setting, providing also a sound and complete constraint based discovery algorithm.

**Strengths:**

1) Paper tackles a novel problem in equivalence class characterization when interventions and selection variables are involved. Authors demonstrate how post interventional selection could give different invariance signatures of p(effect), p(cause) and p (effect|cause) across interventions with and without selection.

2) The characterization and sound and complete algorithms are very novel and elegant contributions as well.

3) Reasonable experimental demonstration that this works both with synthetic data and real world datasets.

**Weaknesses:**

I have only one principal concern - it is with respect to the definition of the Markov equivalence. It is defined only with respect to marginal invariance, conditional invariance between observational distribution and interventional distributions.

 However, it is possible to incorporate invariances between two interventional distributions (not involving the observational one) as in Kocaoglu et.al. 2019 by including augmented variables, one for each pair of interventional target sets. This is possible when a soft intervention on a node could makes the same causal mechanism change irrespective of the interventional collection it is in. For some reason this has not been incorporated in the current set up. This means even with respect to invariance tests, there is a bit more leeway in terms of constraining the equivalence class.
rct64e3

**Questions:**

1) Can authors explain how one would extend to the setting where invariances across two interventional sets can be incorporated ?

2)  Why is the performance curves in the synthetic data very close to FCI-Interven (FIgure 6) ? Is it that latent variable presence dominates the selection variables effect ? Is it possible to introduce more selection variables than latents to see experimentally if the separation is bigger ?

3) Is there a typo in Line 357 where two squares case needs to be denoted $\\square - \\square$ with an edge in between ?

---

> ### Author Response · Authors · 2025-11-20
> **Response to Reviewer uvY9 (Part 1/2)**
>
> We sincerely thank you for the time dedicated to reviewing our paper and valuable feedback. Please find the response to your comments and questions below.
>
> **Q1:** Can authors explain how one would extend to the setting where invariances across two interventional sets can be incorporated?
>
> **A1:** **Invariances between two interventional sets can be incorporated by augmenting our framework with the exogenous binary variable $\psi$ (pair-specific “F-nodes” in the sense of Kocaoglu et al.), as mentioned in Step 3 of our $\mathcal{F}$-FCI algorithm to incorporate the rules proposed by Kocaoglu et al.. In principle, the priority of $\mathcal{F}$-FCI is to systematically leverage invariance\variability patterns between different intervention settings to identify the structure among intervened variables, irrespective of the source of change.** The observation–intervention case is a special instance of the intervention–intervention setting, and the two sources of distributional change are orthogonal. Consequently, if cross‐setting changes can be isolated (e.g., via appropriate conditioning or by contrasting a single‐variable intervention with observation), both sources can be exploited to further orient the graph by including an exogenous intervention indicator $\psi$ (cf. Rule 9 in Kocaoglu et al.). At the same time, although both approaches utilize changes in different distributions (environments),  $\mathcal{F}$-FCI  can handle post-treatment selection and characterize fine-grained interventional Markov equivalence classes. Specifically,
>
> -   **More general setting.** The approach of Kocaoglu et al. does not address **post-treatment selection**,  making it still under the umbrella of FCI in guarantees for the soundness and completeness of tail and arrowhead marks. In contrast, $\mathcal{F}$-FCI models post-treatment selection explicitly and uses the CI-patterns (Figure 4) to distinguish causation from selection effects in complex structures, yielding fine-grained interventional Markov equivalence classes.
>
> -   **Orienting more edges.** The exogenous binary variable represents two settings, either observation vs. intervention or two distinct interventions. $\mathcal{F}$-FCI additionally **orients edges between differently intervened variables**, whereas Kocaoglu et al.’s Rule 9 primarily orients **between intervened and observed variables**.
>
> ---
>
> **Q2:**  Why are the performance curves in the synthetic data very close to FCI-Interven (Figure 6)? Is it that the latent variable presence dominates the selection variables' effect?
>
> **A2:** The performance of $\mathcal{F}$-FCI and FCI-INTERVEN is acutally not very close. Although the curves may look close because the y-axis spans 0.2-0.8 across all methods, $\mathcal{F}$-FCI achieves around **10% improvement in DAG Precision** compared with FCI-INTERVEN on average in Figure 6. This gain stems from its superiority not only in handling post-treatment selection, but also in identifying causal relations and selection structures in complex settings (motivation examples in Figure 1).
>
> Moreover, this improvement is **not** due to the dominance of latent confounding. In our synthetic generator, **both latent confounders and post-treatment selection are present**; each instance includes **2–3 selection variables** and **2–3 latent confounders** (Line 454). The observed gains therefore reflect $\mathcal{F}$-FCI’s ability to explicitly handle post-treatment selection and to exploit intervention-induced invariance/variability to characterize **Fine-grained interventional Markov equivalence classes** (i.e., recover more concrete causal structure), rather than the imbalance in latent confounders.

---

> > ### Author Response · Authors · 2025-11-20
> > **Response to Reviewer uvY9 (Part 2/2)**
> >
> > **Q3:** Is it possible to introduce more selection variables than latents to see experimentally if the separation is bigger?
> >
> > **A3:** Yes. Following your suggestion, we conduct additional experiments on synthetic data with **5-6 randomly sampled selection variables** per instance. The results show that $\mathcal{F}$-FCI achieves **larger gains (more than 15% in Precision for both hard and soft intervention) over FCI-INTERVEN** with the number of post-treatment selections increases, confirming $\mathcal{F}$-FCI’s effectiveness in explicitly handling post-treatment selection.
> >
> > Experimental setting: hard intervention, 20 variables, 2-3 latent confounders, 5-6 selection variables.
> >
> > | | **$\mathcal{F}$-FCI** | | | | **FCI-INTERVEN** | | | |
> > |---|---|---|---|---|---|---|---|---|
> > |  n | Precision | SHD | F1 | Recall | Precision | SHD | F1 | Recall |
> > | 500 | 60.1 ± 0.6 | 12.6 ± 1.8 | 56.7 ± 0.2 | 55.7 ± 1.0 | 46.7 ± 0.1 | 16.3 ± 3.2 | 54.8 ± 0.1| 69.0 ± 2.1 |
> > 1500 | 62.1 ± 0.6 | 12.4 ± 4.2 | 58.2 ± 0.7 | 57.1 ± 2.1 | 46.8 ± 0.1 | 16.5 ± 3.6 | 56.8 ± 0.2 | 74.5 ± 2.2 |
> > |2000 | 64.2 ± 0.4 | 10.9 ± 4.7 | 64.7 ± 0.5 | 66.4 ± 1.3 | 43.7 ± 0.1 | 17.2 ± 6.7 | 55.2 ± 0.1 | 77.8 ± 2.1 |
> >
> >
> > Experimental setting: soft intervention, 20 variables, 2-3 latent confounders, 5-6 selection variables.
> >
> > | | **$\mathcal{F}$-FCI** | | | | **FCI-INTERVEN** | | | |
> > |---|---|---|---|---|---|---|---|---|
> > |  n | Precision | SHD |  F1 |  Recall | Precision |  SHD |  F1 | Recall |
> > | 500 | 67.6 ± 0.6 | 9.8 ± 3.9 | 63.0 ± 0.8 | 61.6 ± 1.0 | 49.0 ± 0.2 | 14.2 ± 1.6 | 54.1 ± 0.3| 62.2 ± 1.3 |
> > 1500 | 68.7 ± 0.4 | 8.7 ± 3.2 | 69.1 ± 0.4 | 70.2 ± 0.4 | 49.8 ± 0.6 | 14.1 ± 5.49 | 58.3 ± 0.3 | 72.2 ± 1.1 |
> > |2000 | 70.3 ± 1.5 | 8.6 ± 7.2 | 68.8 ± 0.9 | 68.9 ± 0.7 | 48.1 ± 0.4 | 14.4 ± 5.0 | 57.1 ± 0.2 | 72.1 ± 0.6 |
> >
> > ---
> >
> > **Q4:** Is there a typo in Line 357 where the two-square case needs to be denoted □-□ with an edge in between?
> >
> > **A4:** **This is not a typo, Type II inducing node is marked with  □□.** Type II inducing node denotes the middle node on the path whose incident endpoint marks at that node are both □ in $\mathcal{F}$-PAG. For example, considering $X_1 \rightarrow X_2 \rightarrow X_3$, $X_1 \rightarrow S \leftarrow X_2$, $X_2 \leftarrow L \rightarrow X_3$. If the causal structure is identifiable, **the output $\mathcal{F}$-PAG is $X_1-□X_2□\rightarrow X_3, X_1 \rightarrow X_3$, where $X_2$ is the Type II inducing node marked with □□.**

---

### Author Response · Authors · 2025-12-01
**General Remarks**

Dear AC and Reviewers

We thank the AC for your time and contribution to the community, and we are grateful to all reviewers for their thoughtful suggestions and kind recognition of our contribution.

Contributions recognized by **all reviewers** include: highlighting the often-overlooked issue of post-treatment selection, characterizing a novel fine-grained interventional Markov equivalence class,  designing its corresponding graphical representation, and proposing a new algorithm to identify this equivalence class with provable soundness and completeness.

|Concerns & Suggestions|By|Our Response|
| --- | --- | --- |
| Principal concern about two intervention settings|Reviewers UvY9|We have added a detailed description of how the general $\mathcal{F}$-FCI framework can incorporate two intervention settings. We clarify that the main contribution of $\mathcal{F}$-FCI lies in systematically leveraging invariance/variability patterns across different intervention settings to identify the structure among intervened variables, irrespective of the source of change.|
|Further verification for handling post-treatment selection. |Reviewer UvY9| We have incorporated additional experimental results (two tables) to verify the performance of $\mathcal{F}$-FCI in scenarios with more post-treatment selections (5-6).|
| Comparison with pre-selection|Reviewer eoQR| We now include a detailed comparison between pre-treatment and post-treatment selection at the level of definitions (principles), illustrative examples, interaction with interventions, and structural causal model.|
|Robustness of $\mathcal{F}$-FCI in realistic settings|Reviewer 1GEE| We have added experiments with results summarized in six tables to demonstrate robustness under different nonlinearities, various noise distributions, and larger graphs. |
|Case study| Reviewer SifF| We have added a case study with a specific example to highlight the differences between $\mathcal{F}$-FCI and the baselines, clearly illustrating how $\mathcal{F}$-FCI recovers a finer-grained equivalence class compared to existing methods.|
|Clarification of technical terminology|Reviewers UvY9, eoQR, SifF|We have refined and corrected unclear terms previously in the revision.|

Sincerely,

The Authors of Submission 10458

---

### Meta-Review · Area_Chair_HEB4 · 2026-01-05

**Summary:**

This is an excellent theoretical contribution to interventional causal discovery, and given the reviewers' consistent, unanimous, and positive evaluation, I recommend acceptance without hesitation. During the rebuttal, the authors have further strengthened the paper's theoretical framework by providing additional empirical assessment. In their responses, three of the reviewers have noted that they maintain their positive ratings (and the fourth one has not responded).

**Reviewer Concerns:**

Reviewers have collectively commented positively on the importance of the problem and the algorithmic contribution of the paper. They have also raised several questions/concerns: whether the proposed FI-Markov equivalence could be tightened by leveraging invariances across pairs of interventional distributions; whether the empirical results clearly demonstrate gains over strong baselines; and how the work relates to and differs from pre-treatment selection frameworks. The reviewers have also raised a number of issues related to clarity/notations/, and typos.

The authors have directly responded to all the above comments and questions, by clarifying how their framework can incorporate intervention–intervention invariances, providing new experiments, explaining the the average improvements, recognizing the explicitly stated limitations. They have also added a clear comparison to pre-treatment selection (definitions, examples, intervention interaction, SCM differences) and fixed the various notation/definition/typo issues flagged by reviewers.

**Reviewer Scores:**

Three (our of four reviewers) have mentioned that they'd maintain their positive ratings. One reviewer has not provided additional comments after the authors' responses.

---

### Decision · Program_Chairs · 2026-01-26

Accept (Poster)